# Ice-volume-forced erosion of the Chinese Loess Plateau global Quaternary stratotype site

T. Stevens [1], J.-P. Buylaert [2,3], C. Thiel[4], G. Újvári [3,5], S. Yi[6], A.S. Murray [2], M. Frechen[4] & H. Lu[6]

The International Commission on Stratigraphy (ICS) utilises benchmark chronostratigraphies to divide geologic time. The reliability of these records is fundamental to understand past global change. Here we use the most detailed luminescence dating age model yet published to show that the ICS chronology for the Quaternary terrestrial type section at Jingbian, desert marginal Chinese Loess Plateau, is inaccurate. There are large hiatuses and depositional changes expressed across a dynamic gully landform at the site, which demonstrates rapid environmental shifts at the East Asian desert margin. We propose a new independent age model and reconstruct monsoon climate and desert expansion/contraction for the last ~250 ka. Our record demonstrates the dominant influence of ice volume on desert expansion, dust dynamics and sediment preservation, and further shows that East Asian Summer Monsoon (EASM) variation closely matches that of ice volume, but lags insolation by ~5 ka. These observations show that the EASM at the monsoon margin does not respond directly to precessional forcing.

[1] Department of Earth Sciences, Uppsala University, Villavägen 16, 75236 Uppsala, Sweden. [2] Nordic Laboratory for Luminescence Dating, Department of Geosciences, University of Aarhus, DTU-Risø campus, Frederiksborgvej 399, 4000 Roskilde, Denmark. [3] Center for Nuclear Technologies, Technical University of Denmark, DTU-Risø campus, Frederiksborgvej 399, 4000 Roskilde, Denmark. [4] Leibniz Institute for Applied Geophysics, Section S3: Geochronology and Isotope Hydrology, Stilleweg 2, 30655 Hannover, Germany. [5] Institute for Geological and Geochemical Research, MTA Research Centre for Astronomy and Earth Sciences, Budaörsi street 45, Budapest H-1112, Hungary. [6] School of Geography and Ocean Science, Nanjing University, 210023 Nanjing, China. Correspondence and requests for materials should be addressed to J.-P.B. (email: jabu@dtu.dk)

The margins of deserts are highly sensitive to climate change and human influences[1]. Small changes in vegetation, climate or land use drive major changes in sand dune and dust activity[2, 3], which in turn have major impacts on local populations, global dust emissions and climate forcing[4, 5]. By extension, sedimentary records from the margins of deserts are highly sensitive indicators of past environmental change in these crucial areas[6]. The desert margin of the Chinese Loess Plateau (CLP; one of the world's most important terrestrial climate archives) is especially significant in this regard. In addition to recording East Asian Monsoon climate and Asian aeolian dust dynamics in loess and palaeosol units (systems that alter global climate and now affect billions of people), the area also records expansion and contraction of a desert sand sea that has experienced significant Holocene and recent desertification[7–9]. This is of particular relevance, given that the loess–palaeosol climate proxy record from the CLP desert marginal Jingbian site (Fig. 1) has been adopted as the terrestrial stratotype for the International Commission on Stratigraphy (ICS) global benchmark Quaternary chronostratigraphic scheme[10, 11], plotted on the orbitally tuned CHILOPARTS time series[12]. The ICS chart for the Quaternary is the reference point for climatic evolution over the past 2.7 Ma, including marine isotope stages, the Antarctic isotope record, and the CLP and Lake Baikal sequences[13]; it underpins our fundamental understanding of past global environmental change and enables correlation of stratigraphic records worldwide. Its accuracy is therefore of central importance in past climate research.

Nevertheless, our understanding of how processes in desert marginal environments impact the preserved sedimentary record is limited, and the longer-term driving forces behind sand activity remain debated due to the limited preservation potential of dune sediments[9]. Sandy desert areas are known to be highly complex and dynamic environments, with the location of deposition and erosion shifting rapidly and across small distances in response to forcing by winds and precipitation[14]. Jingbian lies in an area that

has been covered by expanded sand dunes in the past[15]. Such processes could therefore severely compromise the completeness of the stratigraphic record and undermine the integrity of correlation based, non-independent chronostratigraphic models such as the one used in the ICS scheme. Furthermore, detailed optically stimulated luminescence (OSL) dating of more central CLP sites over the last glacial has shown that age models derived from correlation-based methods contain significant inaccuracies of up to 10 ka[16]. More fundamentally, a recent proposal argues that the CLP is a highly dynamic environment which leads to substantial internal aeolian recycling of pre-deposited material and a reduction in CLP area size[17, 18]. Such sediment recycling would undermine routine desert marginal CLP palaeoenvironmental reconstruction as well as the basis of understanding of past monsoon, dust and desert dynamics in this region. By implication this hypothesis also calls into question the accuracy of the ICS Jingbian chronostratigraphy. It is thus crucial that the past loess and desert record at Jingbian is independently constrained.

Here we develop a fully independent age model for the Jingbian section over the last ~250 ka using a combination of the quartz OSL and K-feldspar post-IR Infra-Red Stimulated Luminescence (IRSL) techniques[19, 20] applied at high sampling resolution. This model shows that Jingbian is characterised by numerous hiatuses of up to ~60 ka that are highly spatially variable across a heavily eroded gully section (Fig. 2). This radically changes the palaeoclimatic interpretation of the sedimentary sequence preserved at the site, supports a revised model for development of the CLP, provides new insights into East Asian Summer and Winter Monsoon (EASM/EAWM) dynamics, and requires a major revision of the ICS chronostratigraphic scheme for Jingbian.

## Results

### A luminescence-based chronostratigraphy for the past 250 ka.
Our new luminescence age model is based on 220 ages on

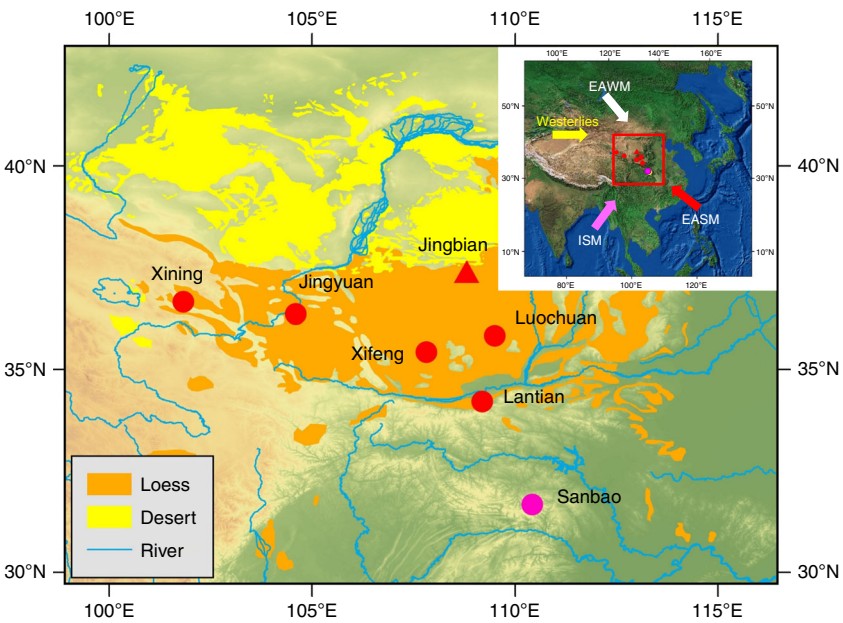

**Fig. 1** Map of Chinese Loess Plateau showing surrounding deserts and rivers and location of the ICS stratotype site Jingbian. The Jingbian site location is marked with a filled red triangle and other well-known loess sites (filled red circles) and the Sanbao cave site (filled pink circle) are also indicated. Inset shows location of map in China and main prevailing wind directions of East Asian Winter Monsoon (EAWM), East Asian Summer Monsoon (EASM) and Indian Summer Monsoon (ISM). The data set is provided by Data Center for Resources and Environmental Sciences, Chinese Academy of Sciences (RESDC) (http://www.resdc.cn). The base map is a coloured DEM map derived from SRTM 90 m data[84] and the inset map is based on http://www.arcgis.com/home/item.html?id=c3265f30461440c2999add34bcae8e0a. A detailed aerial photograph of the Jingbian site with the studied loess profiles (**A**, **B**, **C**, **D**, **E**) marked is given in Supplementary Fig. 1

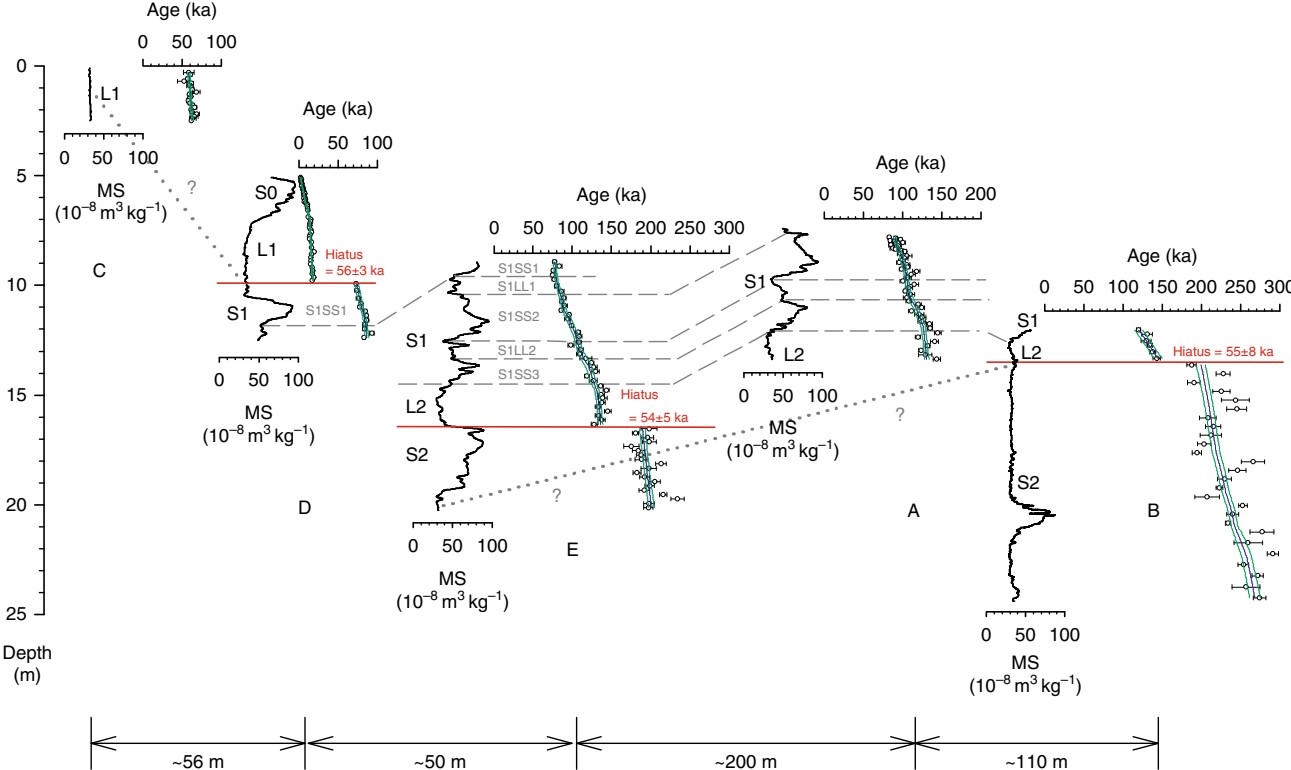

**Fig. 2** Stratigraphic cross-section at Jingbian showing loess–palaeosol stratigraphy and luminescence age–depth relationship for 5 sections. Loess and palaeosol units are denoted with L and S, respectively. All sections start at the modern surface. Section C has the highest elevation and data from other sections are plotted relative to section C. Dashed lines indicate correlations between sections based on low-field magnetic susceptibility and luminescence ages. Discontinuities are shown as horizontal red lines with the length of hiatus quoted. Where the correlation is solely based on luminescence ages and is less certain because of a discontinuity, a dotted line with question mark is used. Ages are plotted with 1 s.d. errors and Bayesian age–depth models are shown as solid blue (weighted mean age) and green (1 s.d. uncertainty envelope) lines. Detailed luminescence age–depth profiles for the individual sections are given in Supplementary Fig. 2a–e

samples taken with a vertical spacing of between 5 and 40 cm at 5 Jingbian sections dug at the ICS stratotype location (Fig. 2). It constitutes the largest and most detailed luminescence data set to date, and to our knowledge is the most comprehensive geo-chronological analysis yet undertaken at a single site. Details on site location, sampling, luminescence dating methodology, age depth modelling and proxy analyses are given in Methods. There are two striking features of the age–depth models for the Jingbian sections (Fig. 2). Large jumps in ages are found in many sections, indicative of large hiatuses in the record of up to 60 ka. Crucially, these substantial gaps are not observable in the visual or proxy stratigraphy and have not been demonstrated previously at Chinese loess sites, yet have major implications for the chronos-tratigraphic model and climate reconstruction. In addition, while the age ranges of some of the sections overlap, the nature of the preserved record at each section is inconsistent, indicating a highly spatially variable relationship between age, depth, sediment type and preservation.

As with many CLP-desert marginal sites, Jingbian is located in a relatively flat plateau landscape with the sections exposed in a deeply incised gully system (Figs. 1 and 2, Supplementary Fig. 1). Some sections show hiatuses where others concurrently show deposition, and yet other sections exhibit extremely high accumulation-rate phases of short duration (Fig. 2). No single section preserves the full sequence covered at the site, as shown in our composite climate records (Fig. 3). As such, these gully sequences require consideration as dynamic landforms, where gully geomor-phology and local morphological context must be taken into account, together with the stratigraphic sequence. One consequence

of this is that while at many CLP sites the Holocene record has been partly disturbed by human activity[21], a uniquely undisturbed 2 m Holocene sequence is preserved at Jingbian (see Fig. 2, section D), protected by unconformable deposition within the gully system and dated by 31 luminescence ages. Thus, the luminescence results reveal that the gully must pre-date the Holocene and that the gully landform itself is recorded in the stratigraphic record at the site. While this dynamism adds to the complexity of interpreting these stratigraphic sequences, our independent dating demonstrates that a detailed composite environmental history can be obtained through luminescence dating of multiple overlapping sections (Fig. 3). This is also reflected in the detailed record of the last interglacial in section E (Fig. 2).

**Ice-volume-forced processes in a desert marginal environment.** When our climate proxy and stratigraphic records are plotted on our new age model against 65°N July insolation[22], marine oxygen isotope stratigraphy LR04 stack[23] and Lake Baikal biogenic silica[13] (Fig. 3), some striking patterns become apparent. Notably, there is a near total lack of preserved record during the last two glacial phases (MIS 2–4 and 6), but with preserved material from the glacial stage MIS 8, as well as interglacials MIS 1, 5 and 7. The large hiatuses appear to terminate close to or following the rapid shift away from peak Northern Hemisphere ice volume at the end of the MIS 2–4 and 6 glacial stages (Terminations I and II). During less positive marine δ18O isotope stages when Northern Hemisphere ice volume was lower, loess sediments are generally preserved. During MIS 7 and the second half of MIS 8, there is relatively low amplitude variability in ice volume and full

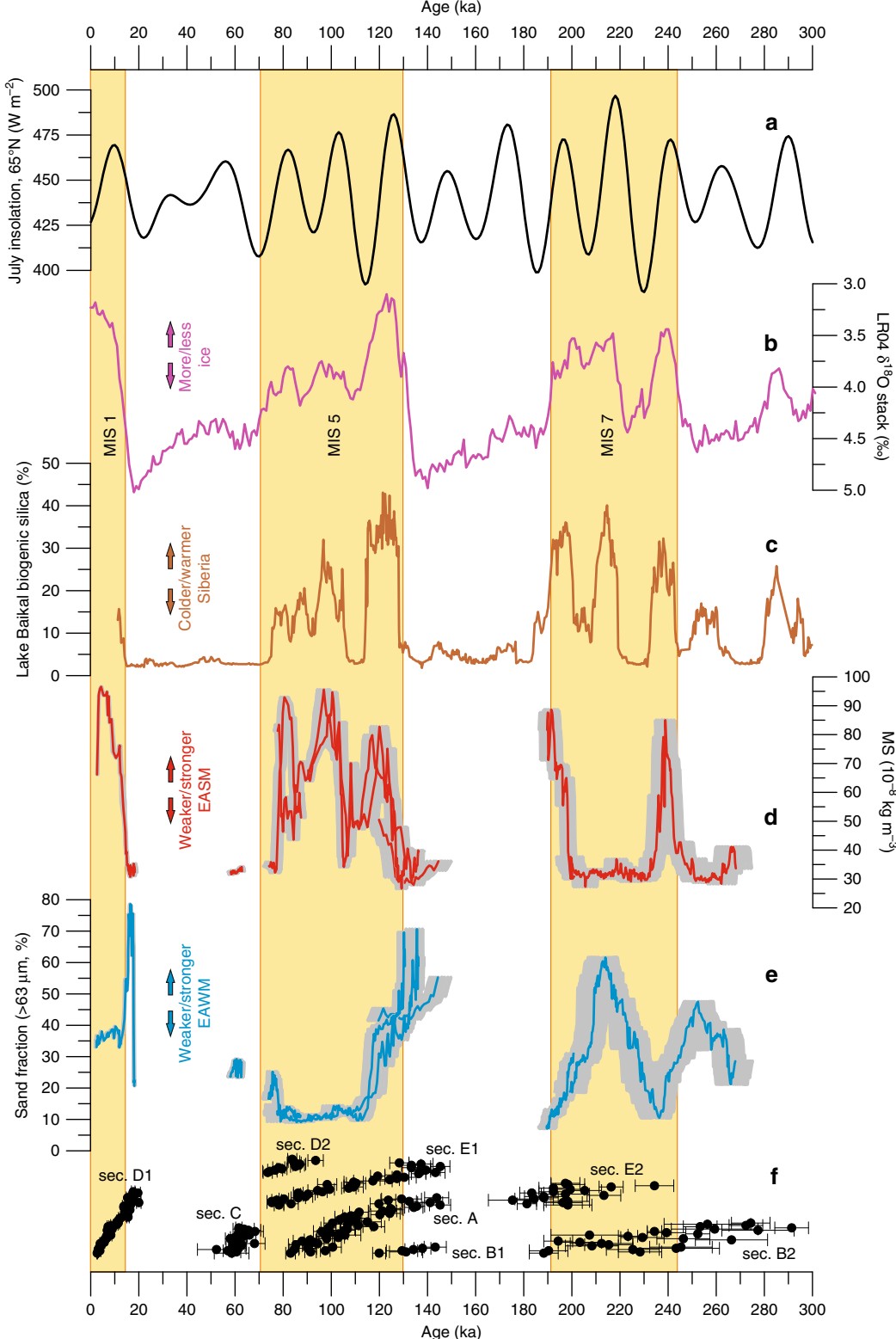

**Fig. 3** Comparison of global/Northern Hemisphere proxy records with the Jingbian records on independent timescales for the last 300 ka. **a** 65°N July insolation[22] record, **b** benthic LR04 δ[18]O stack[23], **c** Lake Baikal biogenic silica[13] record and **d** low-field magnetic susceptibility (MS) and **e** sand content (>63 μm) of the Jingbian loess–paleosol record. Individual luminescence age determinations for each section are shown in **f** with 1 s.d. age uncertainties. Grey shading behind the MS/sand fraction curves indicates 1 s.d. age model uncertainties. Marine oxygen isotope stages/boundaries shown are from Lisiecki and Raymo[23]

preservation of the loess record, including in the comparatively low ice volume glaciation of MIS 8. Two palaeosols associated with the two ice volume minima of MIS 7 are also preserved, separated by a loess unit representing the deep stadial during MIS 7, while MIS 5 and 1, which have no such deep stadials, are only represented as palaeosols at Jingbian.

Based on this pattern and its relationship to the $\delta^{18}O$ record, we propose an explanation of the mechanisms behind desert marginal sediment accumulation, preservation and erosion, and hence the controls on desert dynamics. The greatly enhanced maximal extent of Northern Hemisphere ice limits during peak MIS 2 and 6 is known to have strengthened the Siberian High and moved the polar front southwards, enhancing cold air outbreaks, and strengthening winds and aridity[24]. The associated water stress would have reduced vegetation stabilisation of dunes while cold air outbreaks would have driven seasonally strong winds, promoting deflation and sediment movement; these erosive processes changed Jingbian from a depocentre into a dust source and account for the hiatuses at the site. On shorter timescales, the polar front, modulated by Atlantic Meridional Overturning, has been shown to drive strengthened EAWM circulation and dust deposition on the Loess Plateau[25], while the strength of the Siberian High is tied to ice volume and snow cover over multiple timescales[24, 26], supporting this model. While the MIS 2–4 and 6 hiatuses cover most of these glacials (Fig. 3), accumulation may still have occurred locally, but the strong erosional events during peak glaciation would have removed previously deposited material. Sand was deposited at the end of both hiatuses, indicating both an expansion of the Mu Us desert and some dune stability. As no sand was preserved during the prior glacial episodes, a highly mobile sand sea is implied, close by or covering the site and providing a plentiful supply of saltating impactor grains to promote deflation of existing deposited material. Thus, the two major hiatuses at Jingbian during MIS 2–4 and 6 are interpreted as erosional unconformities resulting from enhanced dune mobility driving erosion of underlying strata. During glacial MIS 8 and the stadials within MIS 7 and 5, glaciation did not extend as far as during MIS 2 and 6 (Fig. 3) and so cold air outbreaks, winter monsoon intensity and aridity was not sufficient to drive such dune expansion and dust deflation.

Our revised age model and resulting sedimentary history has a fundamental impact on the interpretation of the global benchmark record at Jingbian. In traditional Loess Plateau chronostratigraphic models, loess/sand units and palaeosols are considered of glacial and interglacial age, respectively. Here we propose a different model for Jingbian. In our view, palaeosol units are indeed indicative of interglacial phases of enhanced EASM (high magnetic susceptibility; MS) and weaker EAWM (finer grain size). However, rather than representative of glacial phases, loess units in the upper part of Jingbian appear to be mainly associated with stadials within interglacials, during which relatively increased ice volume drives cold air outbreaks, aridity and enhanced EAWM circulation, with associated silt transport and dust trapping at Jingbian. Deep glacial phases are removed from the record due to erosion, and sand units occur over more restricted time intervals, both within interglacials and glacials, with both indicating enhanced dune activity and expansion of the Mu Us (Fig. 3). Although the proxy records show general antiphase behaviour of the EASM with the EAWM (Fig. 3), sand accumulation can also occur even during enhanced summer monsoon conditions (e.g., MIS 5e). This suggests that sediment availability, EAWM/Siberian High driven winter aridity, and cold air outbreaks and enhanced wind strength drive dune mobility, desert expansion and sand deposition at desert marginal sites[15]. This is in contrast to the idea that dune expansion and deposition is controlled by summer monsoon-driven moisture availability[10]. Recent identification of relict dune sediment from the LGM

preserved in isolated frost wedges in the Mu Us[9] confirms intense aeolian activity at this time, but also argues for the domination of net erosion due to high winds and aridity. This explains the lack of dune record from the last glacial in the Mu Us[8] and supports our deep glacial erosional unconformity model. Thus, desert sand dune activity in this part of China is controlled by the intensity of EAWM circulation in Asia, in turn driven by ice volume in the Northern Hemisphere through the Siberian High. Our findings suggest that during peak ice volume phases, this climatically driven erosion in the Mu Us also extended onto the edge of the CLP, driving development of multiple unconformities in one of the global benchmark Quaternary sediment records. This clearly limits the use of Jingbian as a benchmark site for the Quaternary stratigraphic column, and we suggest that a more central CLP site may be more appropriate for use in the Quaternary chronostratigraphic subdivision. Currently, our results demonstrate that the present ICS scheme for Jingbian is incorrect and should be revised.

In addition, our new chronostratigraphic model has a number of significant implications for understanding the CLP, desert sand and atmospheric dust dynamics, as well as monsoon climate. Jingbian lies just south of an escarpment marking the boundary between the Ordos Platform (including the Mu Us desert) and the northern margin of the CLP. Based on the presence of yardangs and wind-gaps cut into Quaternary strata north of this boundary, as well as on loess provenance data, it has recently been proposed that the escarpment has retreated south and east due to wind erosion during peak glacials in a process of 'aeolian cannibalism' of pre-deposited loess material[17, 18]. Our finding that large amounts of sand and dust are eroded during peak glacial conditions at Jingbian supports the reinterpretation of the CLP as a dynamic landform, with deposits undergoing reworking and recycling along the boundary with the Mu Us desert. This is the first direct, independent evidence to support 'aeolian cannibalism' of pre-existing loess[17, 18] alongside reworked Yellow River alluvial sediments[27] as the source of Quaternary loess to the central CLP and may indicate that indeed the CLP is being reduced in size due to peak glacial wind erosion. That this reworking at Jingbian occurs during those glacial phases (the most recent) with greatest ice volume is also consistent with a long-term increase in aeolian dust CLP accumulation rates over the Quaternary[28]. As glacial stage ice volumes increase and cold air surges penetrate further south, generating large, erosive NW to SE tracking dust storms[29, 30] over the Mu Us, Yellow River alluvial platform and northern CLP, material is reworked and incorporated into younger CLP deposits further south. As such, this apparent long-term accumulation rate increase may be more tied to increasing ice volume and loess recycling rather than to changing aridity or dust source alone. Kang et al.[31] and Stevens et al.[32] noted that independently dated central CLP sites show enhanced dust accumulation during the peak of the last glacial (23–19 ka). This general peak in last glacial dust activity coincides with the hiatuses at Jingbian (Fig. 3), and with erosive activity in the Mu Us[9]. We propose that enhanced ice volume may then also be the driver of enhanced Asian dustiness during short phases of the late Quaternary, and erosion of desert marginal loess will likely directly contribute to increased atmospheric dust loading downwind on the central CLP.

**EASM, ice volume and lagged response to insolation forcing.** EASM-driven MS peaks preserved at Jingbian show a remarkable match with reductions in ice volume (Fig. 3). MS also shows variability at the same frequency as precessional cycles in the Northern Hemisphere summer insolation record, but systematically lags behind July insolation at 65°N (Fig. 4). Multiple independent records and models support a role for precessional forcing in driving Asian summer monsoon intensity[33–36] and

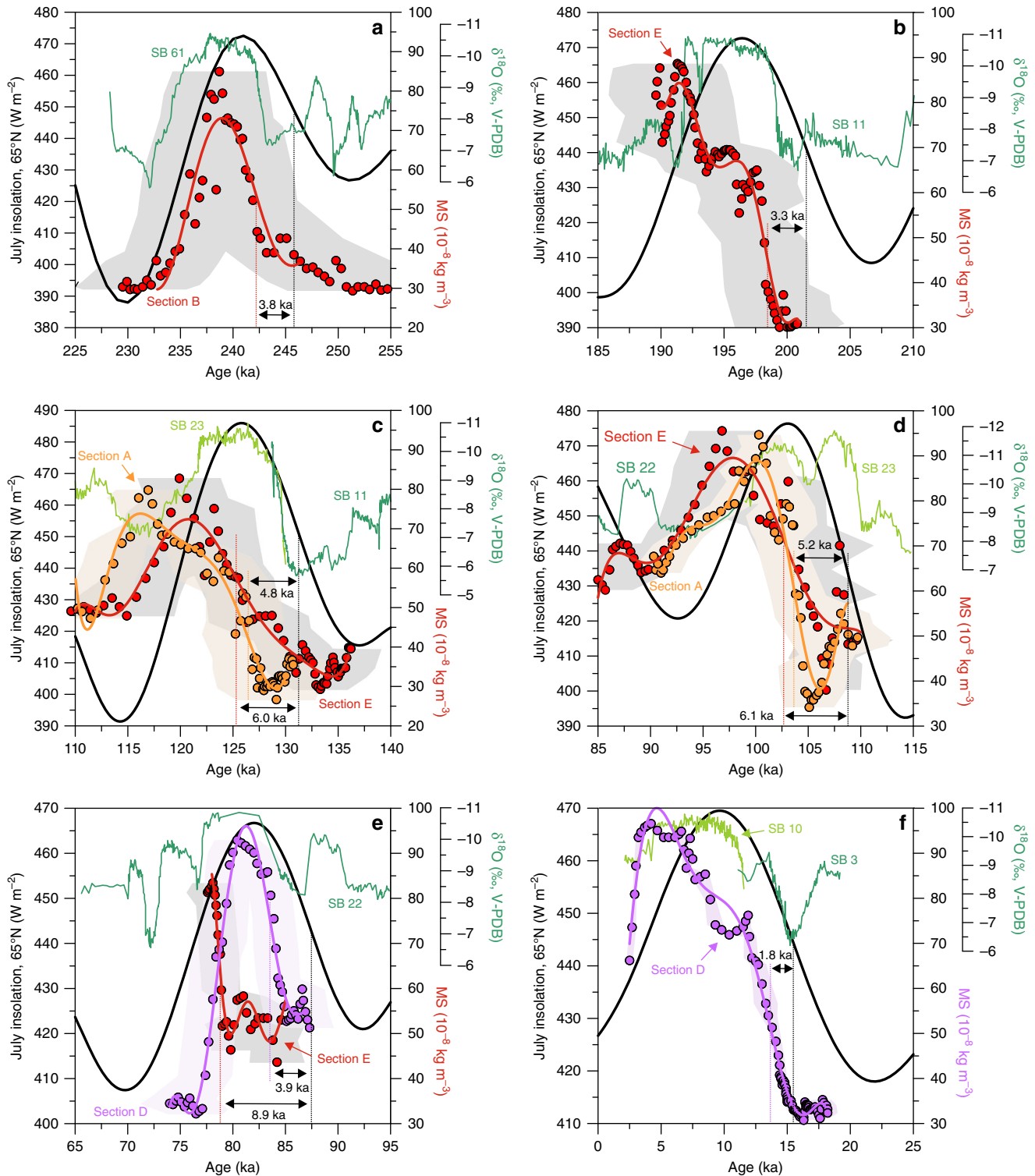

**Fig. 4** Comparison of insolation with East Asian Summer monsoon proxy time series back to ~250 ka. Records shown include 65°N July insolation[22] (black curve), the U-Th-dated Sanbao (SB) cave δ18O records[34, 85] (green lines) and the OSL/pIRIR-dated loess/soil magnetic susceptibility (MS) records at Jingbian (red/orange/mauve symbols and lines) for **a** MIS 8/7, **b** MIS 7/6, **c** MIS 6/5, **d** MIS 5/4, **e** MIS 5 and **f** MIS 2/1 transitions. The red/orange/mauve curves represent fitted polynomials to the MS data sets. Lag calculations for the MS records are given against 65°N July insolation and lag values are specified for the inflection points. Shaded area behind the MS data indicates 1 s.d. age model uncertainties

monsoon variation generally is regarded as a function of low latitude solar insolation[37]. However, over geologic timescales the degree to which there is a direct, singular forcing response of the monsoon to precession, or one where multiple factors such as $CO_2$ and sea level modulate a lagged EASM response, is unclear. Some authors advocate a direct response with no lag, based often on speleothem δ18O records[33, 34], while others argue for a c. 8 ka lag compared to absolute annual maximum insolation, based mainly on marine records[35, 38]. As previous studies only focus on the last glacial termination[39, 40], our results permit the first independently

dated analysis of multiple precessional cycle phase lags between EASM proxies and insolation forcing in the loess record, and provide an independent test of these conflicting hypotheses.

A clear, consistent phase lag between 21st July 65°N isolation and the Jingbian EASM is seen across all transitions in our dataset (Fig. 4, Supplementary Table 2). The insolation lag calculation and the effect of different life-time averaged water content assumptions on this lag are outlined in Methods. While the size varies due to age model uncertainty, the average lag is 4.9 ka (s.e.m. = 0.7 ka, $n = 9$), which would increase to ~7 ka if the target reference curve for phase measurement is taken as the absolute maximum insolation curve, as suggested by Clemens et al[35]. This is within uncertainties of the lag proposed from marine records such as the Arabian Sea summer monsoon stack[41], and contrasts sharply with results from speleothem $\delta^{18}O$[34]. We argue that the observed lag is not related to delays in MS signal acquisition; both theoretical models and empirical evidence point to rapid oxidation/reduction response of iron oxides and formation of superparamagnetic minerals that enhance MS[42–45]. Although transmission of the forcing signal through the climate system may account for some of the lag, we also note that there is remarkable similarity between our independently dated MS record and global ice volume as represented in the LR04 stack[23] (Fig. 3). The only exception is during MIS 7 where a peak in ice volume has no MS/EASM equivalent peak in our Jingbian record (Fig. 3). However this may be an artefact of preservation; the missing peak occurs at the point of increased sand content bracketed by a deeper glacial phase (Fig. 3). Given the larger absolute age uncertainty at this time point and the more scattered ages in the section this data set comes from (B, Fig. 2), it is quite possible an undetected erosional event has removed this peak.

While low latitude insolation directly drives monsoon variability at the precessional band[37], the lagged MS record shows there cannot be a direct response at Jingbian; there must be other factors that heavily modulate the monsoon response in the region. This seems plausible given that the summer monsoon only penetrates as far north as Jingbian due to factors such as land–sea configuration[37]. As such, changes in this configuration due to ice volume would be expected to alter summer monsoon patterns at the site. The match between the MS record and the LR04 stack implies a response of the monsoon at Jingbian to insolation forcing that is similar to the response of the Northern Hemisphere ice sheets, potentially controlled by combined eccentricity, obliquity and precession, or alternatively that Northern Hemisphere ice volume dominates the forcing of the EASM[28, 36]. We suggest that variation in the EASM at Jingbian over the last 250 ka can be explained by combined insolation, ice volume, and $CO_2$ forcing, supported by results from $\delta^{13}C$ of loess organic matter, recent climate model simulations and many marine records[35, 38, 40]. Coupled, ocean–atmosphere–sea ice-land surface climate modelling of the last glacial monsoon suggests that atmospheric $CO_2$ driven high latitude temperature changes drive latitudinal shifts in zonal circulation and the Intertropical Convergence Zone (ITCZ), in turn affecting monsoon precipitation[40]. These shifts would also have affected meridional temperature gradients, snow and ice cover on high ground, ice sheet dynamics, and hence global sea level (land–sea configuration), which in turn will also directly modulate summer monsoon circulation[37, 46–49]. Additional temperature forcing is driven by insolation at high latitude[40]. In monsoon marginal areas like Jingbian, such factors are likely to be the dominant control on summer monsoon dynamics, even if direct precessional forcing dominates monsoon intensity in core monsoon areas[37]. Variations in sea-level and $CO_2$ forcing will alter the spatial extent and coverage of the summer monsoon, which will cause significant changes to precipitation levels at monsoon marginal sites, consistent with our record at Jingbian. As such, the previously

widely accepted hypothesis of dominant direct low-latitude precessional forcing of EASM precipitation patterns seems increasingly implausible at the monsoon margin. Variation in monsoon proxies in various archives is consistent with this geographic effect with regard to monsoon forcing[38], with high latitude forcing exerting a dominant control on monsoon precipitation patterns in monsoon marginal areas. Our data apparently conflict with some speleothem $\delta^{18}O$ records of summer monsoon rainfall[34]. However, reinterpretations of speleothem $\delta^{18}O$ data suggest that either this proxy is not solely influenced by summer monsoon intensity[50] or that $\delta^{18}O$ is a function of integrated rainfall amounts between monsoon source and the cave site[51]. If the latter is true it would imply this integrated rainfall was a function of low latitude precessional forcing. However, this is still consistent with our model as we would expect that integrated summer monsoon rainfall prior to precipitation at cave sites close to the southern part of the CLP would be dominated by low latitude precessional forcing, as this rainfall occurs dominantly in the core monsoon region. However, the extent of summer monsoon rainfall closer to the monsoon margins like on the north CLP, would still be dominantly controlled by the spatial extent and coverage of the summer monsoon, itself modulated by ice volume–sea level–$CO_2$ forcing.

## Methods

**Study site.** The study site is located in Jingbian County and comprises five sections (A, B, C, D, E; where >1 m of material was removed to freshly expose the sediment) (see Supplementary Fig. 1 where locations of individual sections are also given). The elevations of the individual sections were measured to within a few cm using differential GPS and our coordinates measured at section A were 37°29′52.8″N, 108°54′14.4″E. It should be noted that these are different to the coordinates given for the Jingbian site by Ding et al.[10]. However, as we outline below, there appears to be an error in the site coordinates quoted in Ding et al.[10] and we here demonstrate that in fact we are working on the same site; the ICS stratotype section. Firstly, coordinates for the stratotype site position subsequently given to us by E. Derbyshire are 37°29′58.74″N and 108°54′2.72″E (E. Derbyshire, personal communication 2015), with an elevation of ~1700 m above sea level (a.s.l.). Note that these coordinates refer to the position of a pylon/mast immediately to the west of the gully and are different to the coordinates given for the site by Ding et al.[10] in which Derbyshire is a co-author. The coordinates provided by Derbyshire are also ~330 m from our differential GPS measured location of section A (see above), consistent with the position of the section on the east side of the gully ~300 m from the pylon (Supplementary Fig. 1). Furthermore, Ding et al.[52] first presented the Jingbian section, which was subsequently analysed in Ding et al.[10]. Here they noted that the section was located near the settlement of Guojialiang. Indeed, the nearest settlement to both our sampling site and the revised coordinates provided by Derbyshire is Guojialiang. However, the coordinates given in Ding et al.[10] provide a location ~40 km from Guojialiang, inside the Mu Us desert sand field, with this location also inconsistent with the site descriptions given in Ding et al.[10, 52] and lacking any obvious gully exposure. Finally, during our fieldwork in, a local farmer confirmed that a group of Chinese scientists had worked previously at our sections D and E, and we could distinguish prior sampling (presumably for grain size and/or MS) at many sections within the gully. We therefore conclude that the site coordinates given in Ding et al.[10] are erroneous. Given the revised coordinates from Derbyshire and the match of our sections with the site descriptions and nearby settlements in Ding et al.[10, 52], we are very confident that we were working at the same site as is described in Ding et al.[10] and therefore the ICS stratotype site.

**Luminescence dating.** Samples for luminescence dating were collected by hammering stainless steel tubes (diameter 2.5 or 5 cm; length up to 25 cm) with a vertical spacing of 5–40 cm into freshly cleaned sediment profiles. The tubes were opened under subdued orange light at the Nordic Laboratory for Luminescence Dating (Aarhus University, DTU Risø campus, Denmark). The outer ~5 cm of each tube end was removed and reserved for dose rate analysis (see below). The inner material was wet-sieved to extract the 63–90 and 90–180 μm grain size fractions. These fractions were treated with HCl and $H_2O_2$ to remove carbonates and organic material, respectively. The fractions were etched for 20 min in 10% HF to remove coatings and the outer alpha irradiated layer. After washing in 10% HCl, the fractions were dried and quartz and K-feldspar rich extracts (K content = 12.70 ± 0.10%, $n = 5$) were separated using a heavy liquid solution (LST 'Fastfloat') with density 2.58 g cm$^{-3}$. For the samples from the D section, the quartz-rich fraction was subjected to concentrated HF treatment for 60 min to remove any remaining feldspar. The purity of the quartz OSL signal was confirmed by the absence of a significant IRSL signal using the OSL IR depletion ratio[53]. Both quartz and K-

feldspar rich fractions were mounted as multi-grain aliquots containing hundreds of grains on stainless steel cups.

All luminescence measurements were carried out using Risø TL/OSL DA-20 luminescence readers equipped with calibrated $^{90}Sr/^{90}Y$ beta sources delivering between ~0.10 and ~0.20 Gy s$^{-1}$ to multi-grain aliquots in stainless steel cups. Quartz grains were stimulated using blue LEDs (470 nm; ~80 mW cm$^2$) and the OSL signal was detected through 7.5 mm of U-340 glass filter. Feldspar grains were stimulated using IR LEDs (870 nm; ~140 mW cm$^{-2}$) with the IRSL signal detected through a blue filter pack (combination of 2 mm BG-39 and 4 mm CN-7-59 glass filters). Single aliquot regenerative-dose (SAR) protocols[54] were used to determine the quartz OSL and K-feldspar post-IR IRSL equivalent doses (Supplementary Table 1).

For the quartz measurements, a preheat of 260 °C (duration: 10 s) and cut-heat to 220 °C was used; each SAR cycle ended with a high temperature (280 °C) blue light stimulation for 40 s. Natural, regenerative and test dose signals were measured at 125 °C for 40 s. The initial 0.00–0.32 s of the signal minus an early background (0.32–0.64 s) was used for dose calculation. Feldspar aliquots were preheated at 320 °C for 60 s for natural, regenerative and test dose signals. They were then stimulated twice with infra-red light for 200 s. The first IR stimulation temperature was 200 °C (IR signal) and the subsequent IR stimulation temperature was 290 °C (post-IR IRSL signal, pIRIR$_{200,290}$). The IR clean-out at the end of each SAR cycle was carried out at 325 °C for 200 s. The first 2 s of the post-IR IRSL signal minus a background estimated from the last 50 s was used for dose calculation.

It is well-known that quartz OSL from Chinese loess is dominated by the fast component and generally behaves well in a SAR protocol[32, 55–57]. However, typically age underestimation is observed when doses >~150 Gy are measured in loess using quartz SAR OSL[56, 58, 59]. Therefore, we restricted the use of the quartz OSL signal to samples from the upper 480 cm in section D. Below this limit the quartz SAR OSL $D_e$ values are ≥160 Gy and these results were not used for age depth modelling. Figure 5a, b shows the results of a preheat plateau test on sample D38141. It can be seen that over a wide temperature interval quartz $D_e$ is independent of preheat temperature, recycling ratio is close to unity and recuperation is low. A dose recovery test[60] using the SAR protocol outlined in Supplementary Table 1a was carried out on 10 samples from section D (D38102, −04, −10, −13, −18, −24, −40, −54; 6 aliquots per sample) with given doses ranging between 10 and 50 Gy. Prior to giving the laboratory dose, the natural quartz OSL signal was reset by two blue light stimulations (100 s each) separated by a 10,000 s pause to allow any photo-transferred charge in the 110 °C TL trap to decay. The results of the dose recovery test are shown as a histogram and a measured to given dose plot in Fig. 5c, d, respectively. It can be seen from these results that our SAR protocol (preheat 260 °C/10 s, cut-heat 220 °C) is able to measure a quartz dose given prior to any heat treatment with an acceptable degree of accuracy.

Figure 6a, b illustrates the relationship between the individual aliquot $D_e$ values (normalised to the sample mean $D_e$) and the recycling ratio and the OSL IR depletion ratio for 54 samples from section D. There does not appear to be any trend in these relationships indicating that the $D_e$ value cannot be improved by

rejecting aliquots with relatively poor recycling ratios (e.g., deviating >10% from unity) and that the quartz $D_e$ values are insensitive to the levels of feldspar contamination present in these extracts. The quartz $D_e$ values for section D are tabulated in Supplementary Data 1.

Since the discovery of more stable post-IR IRSL signals[61] compared to conventional IRSL signals measured at ambient temperature, several SAR protocols have been developed to use IRSL to date beyond the quartz OSL dating range[19, 62–64]. Section A of the Jingbian site has already been dated using IR stimulation at 290 °C after IR stimulation at 200 °C (i.e., pIRIR$_{200,290}$; Supplementary Table 1b)[20]. Here we present more laboratory tests of the pIRIR$_{200,290}$ signal from the coarse-grained feldspar extracts.

Figure 7a shows a first IR stimulation plateau[19] and a multi-elevated temperature (MET) $D_e$ plateau (using the protocol described by Li and Li[64]) for the deepest sample in section B. The first IR stimulation plateau results suggest that an apparently stable pIRIR signal is reached when the first IR stimulation temperature is ≥170 °C. This is consistent with the observations of Li and Li[65] who showed that for Chinese loess samples with $D_e$ values of >~400 Gy, the pIRIR$_{200,290}$ $D_e$ values are greater than pIRIR$_{50,290}$ $D_e$ values. Unfortunately, we did not observe a plateau region in the MET-pIRIR data from this sample and this protocol was not considered further. Based on the first IR stimulation plateau, we chose the pIRIR$_{200,290}$ signal as the preferred dating signal for this study. Three other Chinese loess sections have also been successfully dated using the pIRIR$_{200,290}$ signal from polymineral coarse silt grains[66] and from sand-sized K-rich feldspar[67, 68].

Based on extensive laboratory testing, Yi et al.[68] concluded that in pIRIR dating, it is advisable to check for the dependence of the results on test dose size. Figure 7b presents the dependence of the dose recovery result on test dose size for sample D38135 (sample also used in Buylaert et al.[20]). The dose recovery test was carried out by adding beta doses on top of the natural dose in the sample. From these data we deduce that small test doses (<~20% of the dose to be measured) should not be used when large (>500 Gy) doses are measured, in agreement with the observations of Yi et al.[68]. Colarossi et al.[69] have shown that in their sample at least part of this effect could be attributed to charge carry-over from $L_x$ to $T_x$. Figure 7c presents another dose recovery test on bleached (24 h in Hönle SOL2 lamp) 90–125 μm feldspar-rich grains from sample D38146 (test dose was ~40% of dose of interest). The residual dose in this sample after bleaching was 9.9 ± 0.2 Gy (n = 3) and this value was subtracted from the measured doses. It can be seen that for doses up to at least ~800 Gy, our chosen SAR pIRIR$_{200,290}$ protocol is able to satisfactorily recover a dose given in the laboratory. Based on these results, the test dose size for all our $D_e$ measurements was kept between ~30% and ~70% of the measured dose.

Post-IR IRSL signals bleach at a much slower rate than the quartz OSL signal[19] and there appears to be a residual very-hard-to-bleach (or un-bleachable) component present in the pIRIR$_{200,290}$ signal which needs to be taken into account[70, 71]. Based on a long-term (>80 days) bleaching experiment, Yi et al.[68] concluded that a constant (or very difficult to bleach) residual pIRIR$_{50,290}$ signal amounting to ~6 Gy is reached after bleaching for ~300 h in a Hönle SOL2 solar simulator with a lamp-sample distance of 80 cm. Even though a residual dose of

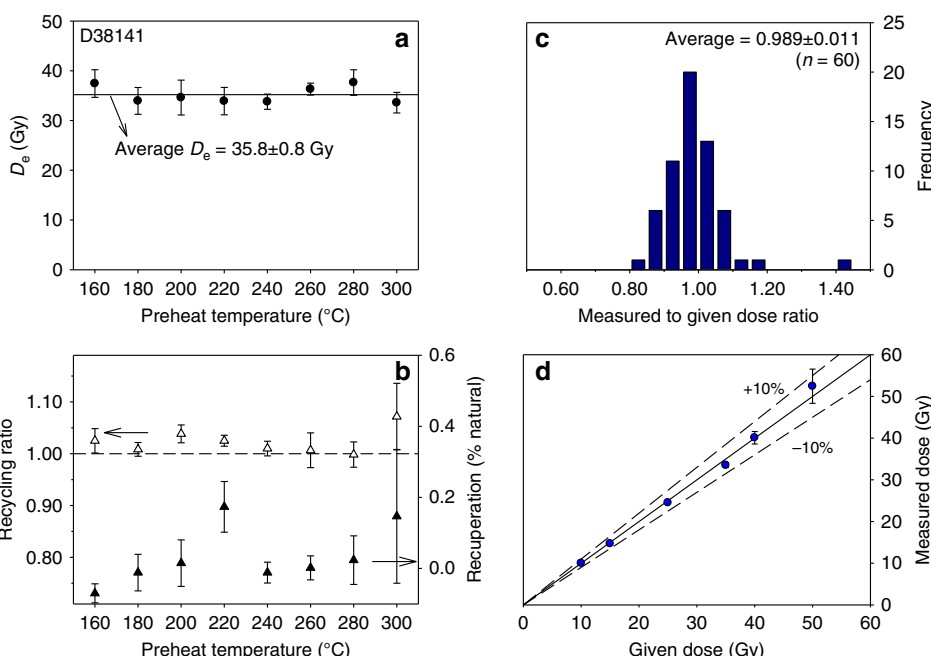

**Fig. 5** Quartz OSL characteristics. **a** and **b** Preheat plateau test for sample D38141; each data point represents the average of six measurements. Dashed line drawn at unity in **b** serves as an eye-guide to illustrate perfect recycling. Results of dose recovery test shown as histogram (**c**) and same data as measured to given dose plot (**d**) for 10 samples from section D. All error bars represent 1 s.e.m.

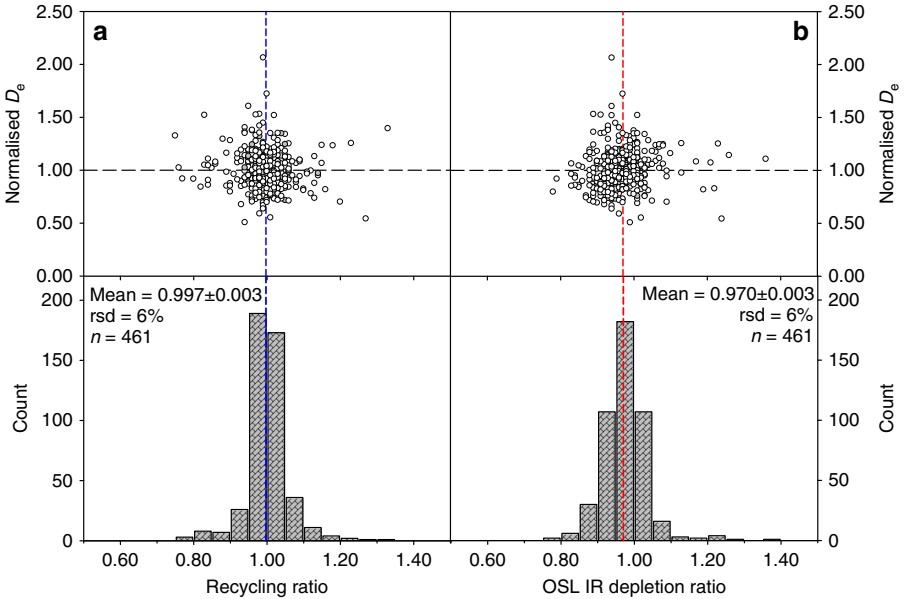

**Fig. 6** Effect of recycling ratio and OSL IR depletion ratio on quartz OSL $D_e$. Samples D38101 to −54 are from section D and at least six aliquots were measured per sample. Bottom graphs show the recycling and OSL IR depletion ratios as a histogram with mean recycling and OSL IR depletion ratios plotted as blue and red dashed lines, respectively. Top graphs show, for the same aliquots, the normalised $D_e$ (to mean sample $D_e$ given in Supplementary Data 1) plotted as a function of recycling ratio (**a**) or OSL IR depletion ratio (**b**). Black dashed lines at unity serve as an eye guide

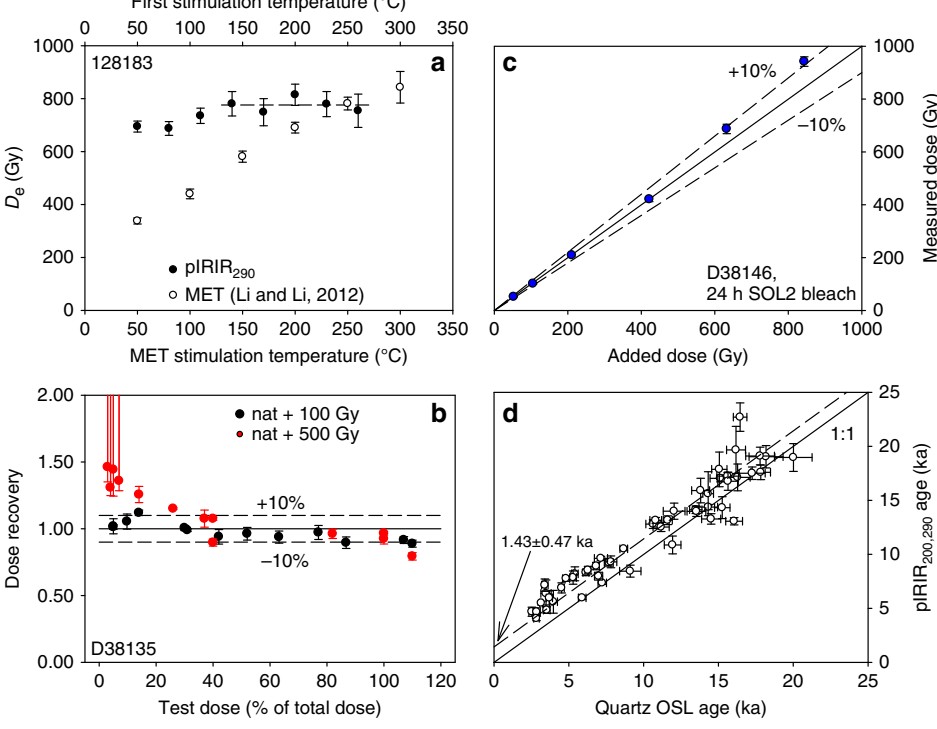

**Fig. 7** Feldspar post-IR IRSL characteristics. **a** First IR stimulation pIRIR$_{290}$ and MET-$D_e$ plateau tests for sample 128183 (section B). **b** Natural+beta dose recovery test for pIRIR$_{200,290}$ signal as a function of test dose size for sample D38135 (pIRIR$_{200,290}$ $D_e$ = 47.8 ± 0.6 Gy, no residual subtracted). Red error bars at low test dose are infinite (saturation). **c** Dose recovery test for pIRIR$_{200,290}$ signal using 24 h SOL2 bleached aliquots of sample D38146 (residual dose of 9.9 ± 0.2 Gy was subtracted from measured doses). Test dose varied systematically between sets of aliquots so that it always was between 40% and 80% of the measured dose. **d** Plot of pIRIR$_{200,290}$ ages against quartz OSL ages for upper part of section D; the data are consistent with a straight line of unit slope and intercept of 1.43 ± 0.47 ka. Error bars in all graphs represent 1 s.e.m.

5–10 Gy in our samples only makes up between 3% and 5% of the lowest pIRIR$_{200,290}$ $D_e$ in our pIRIR$_{200,290}$ age data set (sample D38155), we have estimated this component by comparing the feldspar ages of the upper 450 cm in section D ($n$ = 50) with young quartz OSL ages. This is because it has been shown that fast component dominated quartz OSL signals can record very small doses[72] and for

loess the residual quartz OSL dose at deposition has been shown to be negligible[70]. From Fig. 7d, can be seen that there is overall good agreement between feldspar pIRIR$_{200,290}$ and quartz OSL ages but that there is a small pIRIR$_{200,290}$ offset of 1.43 ± 0.47 ka. Translating this age offset into dose using the average feldspar dose rate

of 3.5 Gy ka$^{-1}$ gives a residual dose of 5 ± 2 Gy. This dose was subtracted from all the pIRIR$_{200,290}$ $D_e$ values prior to calculation of the age (Supplementary Data 1).

Material from the outer end of the tubes was used for dose rate analysis. Samples were first ignited at 450 °C for 24 h, homogenised using a ring-grinder and finally cast in wax in a cup or disc geometry. After storage for >21 days to allow $^{222}$Rn to build up to equilibrium with its parent $^{226}$Ra, they were counted for at least 24 h on one of the six gamma spectrometers from the Nordic Laboratory for Luminescence Dating (Aarhus University). The calibration of the spectrometers is described in Murray et al.[73]. The resulting $^{238}$U, $^{226}$Ra, $^{232}$Th and $^{40}$K concentrations are given in Supplementary Data 1. Note that for some analyses, the data for $^{238}$U is not available due to limited sensitivity of some detectors; in this case the $^{226}$Ra value was used for the entire U series. Radionuclide concentrations were converted into dry beta and gamma dose rates using the conversion factors of Guérin et al.[74]. During calculation of the infinite matrix dry dose rate, we assumed a $^{222}$Rn retention factor of 0.80 ± 0.10 for the $^{238}$U chain; at two standard deviations, this covers a range from no Rn loss to 40% Rn loss. Total dose rates were calculated using life-time average water contents of 10 ± 5 and 15 ± 5% (weight water/dry sediment weight) for loess and soil units, respectively (this assumption is discussed in more detail below). A small cosmic ray contribution to the dose rate was added based on Prescott and Hutton[75].

For K-feldspar grains, we have added an internal beta dose rate based on a K concentration of the feldspar grains of 12.5 ± 0.5%[76]. This assumption has been tested by measuring the K concentration in 5 feldspar rich extracts (one from each section) using an XRF-attachment mounted on the Risø TL/OSL reader. After chemical separation, we are confident that our samples are almost entirely made up of quartz and feldspar. Thus, the XRF instrument is calibrated using a set of standards which are notionally identical, in terms of composition, to end members of the alkali- and plagioclase feldspar series and to quartz; these standards are arranged to fully cover the sample area. This allows us to convert our count rates under the Na, K and Ca X-ray peaks into relative feldspar contributions (i.e. % of total made up of K-feldspar, etc.). The calibration further allows us to attribute a proportion of Si counts to the 3 feldspar contributions, and any remaining Si counts are attributed to quartz. In general, the sum of the 4 components will be less than unity because the sample area may not be fully covered, and so all contributions are normalised to 100%. Once the feldspar analyses have been located on the ternary, the results can be converted to absolute concentrations of K (and Na and Ca if desired) using stoichiometry. The average K content of these five samples is 12.70 ± 0.10%, in excellent agreement with the value proposed by Huntley and Baril[76] (Fig. 8). A Rb concentration of 400 ± 100 ppm was also assumed[77]. There is furthermore a small contribution from U and Th in K-feldspar grains[78] and so an assumed effective internal alpha dose rate contribution from U and Th of 0.06 ± 0.03 Gy ka$^{-1}$ was also included. A lower internal alpha dose rate contribution of 0.02 ± 0.01 Gy ka$^{-1}$ was assumed for quartz grains based on the work by Vandenberghe et al.[79].

The $D_e$ values, radionuclide concentrations, total dose rates and resulting quartz OSL and feldspar pIRIR$_{200,290}$ luminescence ages are given in Supplementary Data 1.

**Age-depth modelling.** Bayesian age-depth modelling was performed using the Bacon code[80], based on altogether 220 OSL/pIRIR$_{200,290}$ data points in sections A–E. Inverse accumulation rates (sedimentation times, yr cm$^{-1}$) were estimated from 3 to 8.8 million Markov Chain Monte Carlo (MCMC) iterations and these

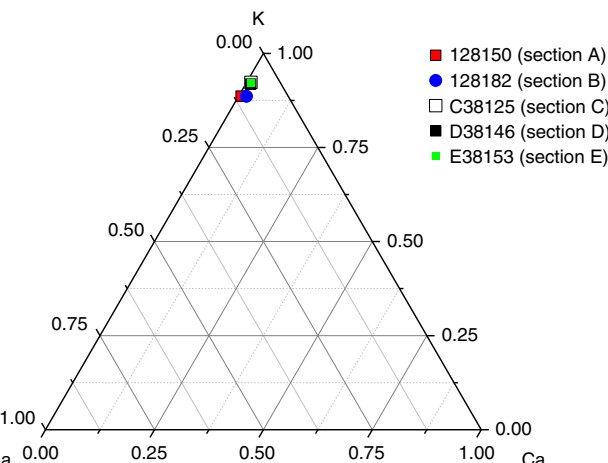

**Fig. 8** Ternary diagram showing XRF analyses of five K-rich feldspar extracts from Jingbian. Sample averages are based on the results from three-to-six aliquots (each aliquot measured three times). Grain size of K-feldspar extracts was 90–180 μm, except for sample D38146 (90–125 μm)

rates formed the age–depth models for each section from A to E (see Supplementary Data 1 and Supplementary Fig. 2a–e). Inverse accumulation rates were constrained by non-default prior information: acc.shape = 1.5 and acc.mean = 0.025–1.0 for the gamma distribution, and mem.mean = 0.7 and mem.strength = 4 for the beta distribution describing the memory effects (or autocorrelation) of inverse accumulation rates. In all cases, the modelling thickness was specified as 20 cm and Gaussian error distributions were applied (i.e., normal = TRUE). Age modelling was run to achieve 5 cm final resolution.

**Proxy analyses.** Adjacent to the luminescence sampling tubes, samples for MS and grain size analysis were collected at 5–10 cm depth intervals. MS samples were measured in the laboratory using a Bartington MS2 magnetic susceptibility metre. Approximately 10 g of each sample was oven-dried at 38 °C, placed into weakly magnetic plastic boxes and measured three times to obtain an average value. Finally, these average values were normalised by the sample mass in order to obtain the mass-specific MS. Grain-size samples were always collected at 5 cm intervals; about 0.2–0.3 g of bulk material was measured using a Beckman Coulter LS13320 laser diffractometer. The samples were dispersed in 1% ammonium hydroxide for 24 h, and sonication was employed immediately prior to adding the sample to the water column. The settings were verified by means of reproducibility tests of more than 50 sub-samples on both soil and loess layers. Five sub-measurements were conducted and at least three sub-measurements were used for averaging. Dismissal of sub-measurements from the averaging was employed when individual curve data implied bubbles in the system. The low-frequency (470 Hz) MS and sand fraction (>63 μm) results are summarised in Supplementary Data 1.

**Lag calculation and water content assumption.** To compare the July insolation curve (65°N) with the MS records at Jingbian, polynomials were fitted to the data sets with an output resolution of 0.1 ka. Minimum values of the first derivative of the polynomials defined inflection points. Lags of the MS records compared to the insolation curve were calculated as age differences between the respective inflection points (Supplementary Table 2).

The effect on the insolation lag of different life-time averaged water content assumptions in luminescence dating is important in this study. Our choice of water content and its uncertainty is first discussed with respect to literature values and the relevance to individual samples is then considered using the section containing the Holocene soil (section D). We then investigate the dependence on different water content assumptions of the apparent lag between our luminescence dated MS record and the insolation record.

Firstly, although there is some variability in the published water content values for Chinese loess (see discussion in Stevens et al.[81]), the values used in this study, of 15 ± 5% w.c. for soil and 10 ± 5% w.c. for loess layers, are similar to previous water content assessments for loess/palaeosols from sites in the N and NW of the CLP[57, 82]. In addition, Chen et al.[83] used a value of 10 ± 5% for a single sample collected in the S8 palaeosol at Jingbian.

We next consider the water content required to reduce the EASM lag to 0 ka for section D. For the two Holocene samples (D38132, w.c. 15% and D38136, w.c. 10%), this would require increasing the water content to ~30% and ~25%, respectively. These water contents are 3 standard deviations from the values used and are close to saturation for sandy loess deposits. However, it is likely that the upper loess–palaeosol units at Jingbian have been well-drained since deposition: the gully is at least 10 ka old since the Holocene soil is inset into the gully system and the current water table is now around 300 m below the sampling level in a >280-m-deep gully system[10]. The river into which the gully flows has incised into Pliocene red clay below the Quaternary loess. The age of this feature is unknown but is likely to be at least multiple glacial–interglacial cycles. It is thus expected that the upper loess–palaeosol units have remained at least several tens of metres above the water table for the majority and probably all of their burial life-time. Thus, we consider it unlikely that the life-time average water content of this site could have approached the levels that would be required to reduce the lag to 0 ka. It is also worth noting that the water content values required for a zero lag would exceed almost all published values for even southern CLP sites, where precipitation levels are double those at Jingbian. If, on the other hand, our water content estimates are too high, the dose rates would be higher, the luminescence ages lower and the lag with the insolation larger. Thus, in all likely water content scenarios, there is a significant lag between the EASM recorded in loess and insolation.

If we now make the additional assumption that the underlying mechanisms causing the insolation lag have not varied systematically with time (which ought to be safe given the lack of an obvious systematic change in ice volume and $CO_2$ back in time at insolation inflection points), we would in turn expect the insolation lag to have remained constant within some bounds over the past ~250 ka. This is precisely what is observed in our data (Supplementary Table 2 and black symbols in Fig. 9). However, increasing the water content by one standard deviation (i.e., from 15% to 20% for soil and from 10% to 15% for loess) causes an increase of 4.7% and 4.1% in the quartz and feldspar ages, respectively. Recalculation of the insolation lag using ages based on these higher water contents introduces a negative trend in the insolation lag versus insolation inflection point graph in Fig. 9 (red symbols). Indeed, using these water contents suggests the physically unrealistic scenario that prior to ~130 ka the loess record of monsoon variability formed before the change occurred in the driving force (change in insolation). A similar

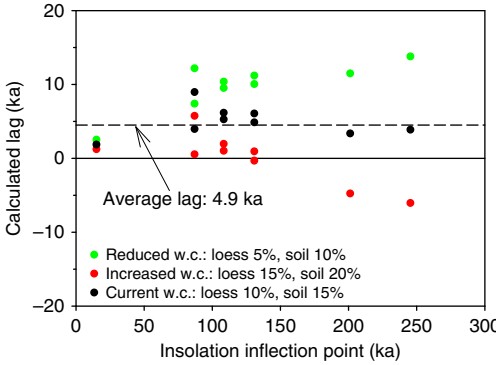

**Fig. 9** Insolation lag as a function of insolation inflection point time for different water content assumptions. Dashed line is drawn at the average of the insolation lag data given in Supplementary Table 2 (black symbols) and shown in Fig. 4. Solid line indicates no lag

but positive trend in the size of the lag is observed when the water content is decreased by 5% (green symbols in Fig. 9). In summary, changing the water content by ±5% introduces trends in the lag with time and increases the standard deviation of the lags from the original ~2 ka (Supplementary Table 2) to ~3 ka. We conclude that our current water content assumption remains the most likely. It does not produce any systematic trend in the insolation lag with time and, if anything, the uncertainty on the water content has been overestimated.

**Data availability**. The data that support the findings of this research can be found in Supplementary Data 1 or upon request from the corresponding author.

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

## Acknowledgements

J.-P.B. thanks the Danish Council for Independent Research | Natural Sciences (FNU) for financial support (Steno grant no. 11-104566). This work was partially supported by the European Research Council (ERC) under the European Union's Horizon 2020 research and innovation programme ERC-2014-StG 639904 – RELOS (awarded to J.-P.B.), and by the National Natural Science Foundation of China (41690111, 41472138). Wang Yao is thanked for help with plotting Fig. 1.

## Author contributions

J.-P.B. designed the study with help from H.L., A.S.M., T.S. and C.T. T.S. wrote the manuscript with help from J.-P.B., G.Ú., C.T., A.S.M. and H.L. J.-P.B., S.Y. and C.T. conducted the fieldwork, J.-P.B. and S.Y. contributed luminescence data, while G.Ú. conducted age modelling. C.T. and M.F. provided the grain size data. T.S., J.-P.B., G.Ú. and C.T. undertook the analysis and interpretation of the results.

## Additional information

**Competing interests:** The authors declare no competing interests.

