## [Peer Review File · Nature Communications]

Reviewers' comments:

Reviewer #1 (Remarks to the Author):

My assessment is that this manuscript presents transformative results and conclusions and should be accepted for publication in Nature Communications. Loess Plateau sections are commonly taken, assumed, or otherwise argued to provide continuous sedimentary archives of Quaternary climate change. This manuscript presents by far the largest and most detailed luminescence dating study of the desert marginal, type Jingbian section. Major findings include:

(1) The presence of large (up to 60 ka), depositional hiatuses during the last two glacial phases (MIS 2-4 and 6). This corresponds to times of peak Northern Hemisphere ice volume. It is proposed that this strengthened the Siberian High and thereby promoted deflation. Reworking of desert-marginal Loess Plateau strata during glacials was recently hypothesized but not demonstrated or quantified until this study.

(2) Loess deposits in the section correspond to stadials within interglacials. This challenges the previous assumption that all major loess deposits of the Loess Plateau correspond to glacials. Hence, the study demonstrates that the stratigraphic record of the desert marginal Loess Plateau is very different than Loess Plateau sections located more distal to the desert margin.

(3) East Asian Summer Monsoon variation at the study site closely matches that of ice volume, but lags changes in insolation by ~4.5 ka. This again, underscores the importance of ice volume in influencing monsoon dynamics of the desert marginal Loess Plateau.

The data strongly support the conclusions and hypotheses. This study will be of very broad interest to Quaternary geologists, climatologists, geomorphologists, sedimentologists, etc. It seems that there is a flurry of new interest, ideas, and debate about all aspects of the Loess Plateau- so this manuscript is very timely. This study highlights both major advances as well as that there will be many more to follow. The history of climate change, landscape evolution, eolian deposition and erosion is much more complicated and dynamic (and hence interesting) than previously made out to be.

Minor comments about how the manuscript could be improved:

(1) The highly spatially variable relationship among age, depth, and sediment type and preservation is noted. Some sections near Jingbian record deposition while others record hiatuses. No single section preserves the full sequence as shown in the composite climate record. This is a very intriguing finding. The following sentence is quoted from the manuscript: "As such, these gully sequences require consideration as dynamic landforms, where the morphological context must be taken into account, together with the stratigraphic sequence." The authors may want to consider elaborating on this statement. What kinds of landforms? Wind-parallel linear loess topography has been documented on the desert marginal Loess Plateau. Is there any correlation between the locations of the sections and the locations of modern ridges/troughs/gullies? Or perhaps the stratigraphy records paleo-topography?

(2) Related to (1) above, it might strengthen the manuscript to briefly summarize some of the geomorphic observations which led to the hypothesis that the Loess Plateau is a dynamic landform- could even indicate some on Figure 1. e.g., Jingbian is located near the windward escarpment margin of the Loess Plateau—proposed to have retreated in the windward direction because of wind erosion during peak glacials. It is most prominent to the west, beyond the city of Dingbian, where there are also large yardangs (or yardang-like landforms) sculpted in Quaternary strata, and wind gaps along the crest of the escarpment. These observations may further strengthen/broaden the conclusions and implications of this study.

Sincerely,

Paul Kapp
15 October 2017

Reviewer #2 (Remarks to the Author):

Stevens, Buylaert et al. Nature Comms. Oct 2017
Ice-volume-forced erosion of the Chinese Loess Plateau global Quaternary stratotype site

This is a well-written, data-rich paper, which focuses on the key Quaternary terrestrial type-section of Jingbian, located toward the northern fringes of the Chinese Loess Plateau in a desert-marginal area close to the present day Mu Us and other north-western deserts. The site has been identified by others (International Commission on Stratigraphy) as a suitable terrestrial stratotype; partly because its location makes it potentially sensitive to Quaternary climate change, and also because of the high rates of sediment accumulation over long time periods which appeared to be essentially continuous over the whole of the Quaternary, making it suitable for global chronostratigraphic correlation with other records such as marine sediments and ice cores. Critically, the paper by Stevens, Buylaert et al. provide: 1) detailed numerical chronologic evidence for the nature and timing of sediment accumulation at the site (rather than using chronostratigraphic methods which involves making assumptions about the continuity of the record of sediment accumulation), and 2) this numerical age control reveals several major (up to ~55-60ka duration) hiatus' in sediment accumulation in various exposures at Jingbian, which 3) calls into question the existing ICS chronology for the site (based on correlation) and hence also questions the validity of the site as a global stratotype.

The paper is important because it shows, for the first time, multiple hiatus' of significant duration (up to ~60ka) that have long been suspected by some researchers, but not previously demonstrated so clearly. The significance of the paper is further heightened by identification of these major hiatus' in a site which has hitherto been formally internationally recognized for its implied continuity of sedimentary record; this is of particular significance and highlights the need for numerical chronology (rather than relying on correlation/chronostratigraphy) at other such terrestrial sites elsewhere. This observation alone, made at multiple sections within the Jingbian site, and underpinned by an impressive dataset comprising >200 luminescence ages, is highly likely to influence academic thinking going forward and is compelling evidence of the potential of such records to be of fragmentary nature in at least some settings (-a point which has been hotly contested by some in the field before now). Additionally, the paper moves beyond these important observations and having secured an independent numerical chronology the authors then explore the timing of change recorded in the climate signals preserved within the Jingbian sediments and compare these to other regional or global climate signals; this can only be done due to the high resolution independent numerical chronology that underpins this work. The implications of these findings for the control of desert dynamics/expansion and also monsoon dynamics are explored at this desert-marginal/monsoon-marginal site, and the changes recorded at this site are linked to global ice volume.

I have some very minor comments and suggestions to improve the clarity at some points in the paper (listed below), but overall I find this to be a very well-written paper, underpinned by a significant and impressive body of data, which allows the authors to demonstrate clearly some major issues with taking a chronostratigraphic approach and correlating records (rather than using independent numerical dating evidence), and enables the authors to explore leads/lags and forcing factors and their sensitive desert-marginal/monsoon-marginal site. As such, I find the paper to be scientifically rigorous, containing new ideas and observations of international significance, and hence likely to influence thinking in the field beyond publication, making it well-suited for

consideration by Nature Communications.

Specific Comments on the main manuscript:

Line 44: strongly suggest adding the word 'terrestrial' to read "one of the world's most important TERRESTRIAL climate archives" (I'm thinking of ice-core and marine scientists reading this!)

Line 46: suggest adding a comma after the bracket to help the reader

Line 59: suggest adding a comma after "record is limited" to help the reader

Line 82-83: "This radically changes the palaeoclimatic sequence preserved at the site..." – I'm not sure that the authors really mean this as written?; it's the word 'sequence' which is potentially an issue here as the reader could confuse this with the sedimentary 'sequence', which is obviously not changed – rather, it is obviously the timescale and hence also the palaeoclimatic interpretation and the accumulation rates etc that are all changed.

Line 107: suggest adding a comma after "site" to help the reader

Line 126: "The large hiatuses appear to terminate following...at the end of the MIS 2-4 and 6 glacial stages (Terminations I and II)." – is the word 'following' appropriate or potentially misleading? The end of one hiatus seems essentially coincident with termination I, whilst accumulation after another hiatus resumes before TII? This may just be an issue of phrasing, rather than an issue of interpretation of the data.

Line 130/131: Difficult to comment on "full preservation of the loess record" for MIS 8 when the full stage is not captured in the record shown – however, where sediments are present (ie from ~270 ka onwards) then the records do indeed seem to be preserved. Again, probably more of a phrasing issue which can easily be resolved.

Line 152/153: "Sand was deposited at the end of both hiatuses, indicating both an expansion of the Mu Us desert and some dune stability." Could enhanced/sustained wind competency also potentially be a factor here? Please comment on whether/how one might distinguish between these various factors (sediment availability, sediment trapping, and wind strength)?
Sentence from line 163-166: "I contrast to traditional...we propose a different model for Jingbian." This phrasing is very awkward for the reader; it is difficult to read due to the many clauses that appear before the main points which start-and-finish the sentence (and which are highlighted in the quoted text pulled out at the start of this comment) i.e. the main point is that the authors are proposing a different model to the traditional view, but this gets lost due to the current sentence structure. Suggest rephrasing this important sentence, for clarity.

Sentence on line 175-180: this is a long yet densely packed sentence; suggest that the authors revisit this sentence and check that it conveys their ideas as clearly as possible.

Line 178: "suggesting that sand sediment availability..." – the word 'sand' or 'sediment' is perhaps intended here, but presumably not both?

Sentence on line 202-205: "That this reworking at Jingbian only occurs during those glacial phases (the most recent) with greatest ice volume is consistent with a long-term increase in aeolian accumulation rates over the Quaternary" – I agree with the sentiment of this sentence and suspect that the authors are correct in what they say, but it is difficult to draw this conclusion solely from the data in this paper (which is what the sentence implies as it is current written), because the 2 reworking/hiatus events take place in the only 2 full glacial times identified and dated in the sequence (MIS 2-4 and MIS 6); for MIS 8 and beyond no record is shown beyond ~270 ka. Hence,

it is not possible to comment on the presence or absence of materials from the sedimentary record (i.e. on hiatuses) for stage 8 and earlier because no records are shown for these glacial periods; additionally, there is no independent numerical chronology for such earlier glacial stages (prior to ~270 ka) hence it is difficult to envisage how any hiatus would be identified in such a case.

Line 220: add a hyphen between "EASM" and "driven" to read "EASM-driven peaks preserved in..."

Line 223: "with a consistent lag..." – it's not a 'consistent lag' (which could be taken to imply a constant temporal offset), but it does 'consistently lag behind July insolation' (in the sense that it is always behind rather than in front of or consistent with the timing of July insolation)

Line 228: suggest adding a comma after 'precession' to help the reader

Line 229: suggesting adding a comma after 'response' to help the reader

Line 238: add units (ka) to the figure of "4.9"

Text within and around lines 249-254: could it also be the case that significantly enhanced sand content (whatever the cause of this may be) simply precludes formation of a palaeosol/development of enhanced magnetic susceptibility ?

Line 253: either 'these data' or 'this dataset' (but not 'this data') Sentence around line 238 and line 256 etc regarding changes in the East Asian Summer Monsoon (EASM) record preserved at Jingbian being reported to lag insolation changes by up to ~4.5ka: I suspect that the authors are correct, but it should also be noted that this lag typically falls within the 1-2 sigma uncertainties on the individual ages and/or the age model, typically used to assess whether or not events are truly separated in time. Having said this, on the positive side, there is a consistency in the data in that a lag is always detected in each case, which suggests that (as long as there is no systematic issue in the OSL dating affecting all ages, such as assessment of the water content) then there may well be a genuine lag in the MS record that is being preserved at Jingbian. The authors may wish to address this issue of uncertainties and the ability to discern these apparent lags in the record.

Line 268: add an Oxford comma after 'simulations' to help the reader

Line 296: consider adding a comma after 'CLP' to help the reader

Line 312: separating "K-feldspar rich extracts" isn't necessarily guaranteed from using a single density separation at 2.58 g/cm³ – please state the K content (as %K or %K₂O) here to qualify/demonstrate that the fractions really are "K-feldspar rich" or moderate the text.

Line 316: please define, within the text, what "absence of a significant IRSL signal" is eg was this assessed using an IR-OSL depletion ratio, or on the basis of raw IRSL counts? If the latter, then how does the raw IRSL count translate into an impact (or not) on a blue OSL signal used for dating?

Line 325: state filter thicknesses (mm)

Line 325-326: amend text to reflect the fact that 2 protocols were used (one for quartz, and one for feldspars) – at present it sounds like everything was measured using one single sequence e.g. amend to say "Single aliquot regenerative dose protocols (M&W, 2000) were used..."

Line 337: "325 °C" for how long (s)? Please add to the text as all other stimulation times are quoted here.

Line 559-564: caption could be improved here to help the reader e.g. adding line colours to the caption text where records are mentioned in the text, and also adding individual graph names (a-f)

to the caption where relevant (eg to denote which groups of figs are related to which specific MIS stages named in the caption). Please also state in the caption whether the red data points are the calculated luminescence ages, or whether they are the ages after a Bayesian fit to the whole luminescence dataset at any given section

Fig 3, within the main manuscript: it could potentially be helpful (and very impressive!) to consider adding an indication of the number and location of independently dated luminescence samples to Fig 3 e.g. by showing a dot for each luminescence age generated in a single line just offset above the age-axis of Fig 3, to show the location (in time, rather than in depth) of each luminescence age determination.

It would also be helpful to add some stratigraphic details to Fig 3, or at least to indicate the location of the major loess units and palaeosols as defined (presumably) on the basis of magnetic susceptibility, if not logged/visible in the field (e.g. to support arguments/observations made in the text, such as those ~line 170).

Supplementary Information:

Line 84: please comment on why small test doses should be avoided (e.g. carry-over of charge?) and give refs where appropriate (e.g. Yi et al 2016; Colarossi et al, 2017).

Lines 94-95: please just clarify by stating in the text the bleaching conditions where >80 days and ~300 h are mentioned (eg daylight bleaching? Sol2? etc).

Add a comma to line 97 to make it easier for the reader (eg after the sample name?).

Line ~122 and Fig S4: please state how the K contents are obtained as Kook et al. 2012 is just a conference abstract and doesn't contain these details. The XRF attachment gives a relative assessment of the K, Na, Ca composition (as subsequently expressed in Fig S4); please explain in the SI text how concentrations (as % K or %K₂O) are then derived. Was any other assessment of the K-content of the separates undertaken (eg beta counting of the material used for dating), and if so what values were obtained?

Fig S2: please state in the caption the number of datapoints that underlie each datapoint plotted in Figs a, b and d (eg the mean and standard deviation of 3 datapoints? etc). I think that the data shown in Figs c and d are derived from the same aliquots, in which case it would be good to say this in the figure caption.

Fig S3 caption: the De value quoted in the caption for sample D38135 (47.8 ± 0.6 Gy) is different to the value in the SI text (43 ± 2 Gy) – please amend/explain.

A comment is also made in the caption that the test dose varies by 40-80 % - please state whether this variation in test dose is just relatively random, or whether it is due to using one (or a limited number) of fixed test dose values, or whether it was varied systematically with increased expected De, etc.

Note also that the last line of the caption should read "the data ARE consistent" (or "the dataset is consistent"), not "the data IS consistent"

Table S2: just a comment that it is great to see bulk density reported here for many samples, even though reporting of these values is not strictly necessary for this particular paper – inclusion of these bulk density values for completeness alongside the age information will, however, be extremely helpful to other studies who consult and use this work in the future. The authors are therefore to be commended for including this information within this SI file.

Reviewer #3 (Remarks to the Author):

Review: Stevens et al., Ice-volume-forced erosion of the Chinese Loess Plateau global Quaternary stratotype site

This work presents results from five loess sections ranging from 2.5 m to 12 m, all in close proximity to one another (~200m maximum separation), near 37°29'58.74"N , 108°54'2.72" E. Age-depth relationships of all sections are very highly constrained with very closely spaced OSL dates, providing the capacity for close examination of the continuity of each section, the longest of which reaches ~280 ka BP (into Marine Isotopic Stage 8). All sites were also measured for magnetic susceptibility and grain size of the sand fraction, the two most commonly employed proxies in loess sections, often interpreted to reflect summer and winter monsoon dynamics respectively.

On the basis of these data, the authors address three main topics including:

I. the implications of their results for the integrity of the International Commission on Stratigraphy (ICS) benchmark chronostratigraphic site at Jingbian (37°40'54"N, 108°31'15" E) located ~40 km to the NW of the sections discussed here,

II. the implications of their results on glacial-interglacial depositional and erosional processes and mechanisms operating in this desert-marginal region, and

III. the implications for our understanding of monsoon dynamics relative to forcing mechanisms such as insolation and high latitude ice volume, with a focus on the issue of timing at the precession band.

Comments on these three components are found below, with the acknowledgement that I am not an OSL expert and defer to other expert reviewers in this regard where our comments might differ. Provided that the OSL dates are technically sound, then the data presented are of high quality and appropriate to addressing the topics covered in this manuscript.

I. Implications for the integrity of the ICS benchmark chronostratigraphy

The authors make a strong case for a significant amount of disturbance and reworking, resulting in well-documented hiatuses labeled in figure 2. Beyond these labeled features, the authors interpret the sections as largely intact and continuous. In the case of Section B (Figure S5_B), however, the OSL dates appear to indicate, if taken at face value, some larger-scale age reversals or disturbed section. See, for example the intervals ~500 to 600 cm, 600 to 800 cm, and 800 to 900 cm where age decreases with depth in the section. Might these indicate slumping with possibly repeated section or simply disturbed section? If so, such disturbance may be why the structure in figure 3d does not match that in the Baikal record within MIS 7, unlike the reasonably good match in MIS 5. In any case, the authors point is valid – that this desert-margin location appears prone to strong depositional/erosional dynamics and may not be the strongest choice for locating an ICS benchmark stratigraphic section. Such a general statement, in and of itself, is valid but the finding is then extrapolated to the ICS type section 40 km to the NW with the statement (abstract lines 21-24, title, text lines 194-194) that the chronology at the type section is inaccurate. A statement of this order should be backed up with an in-depth comparison documenting where, in the last 300 kyrs, the chronology of the type section is compromised. For example, the authors document clearly that glacial intervals MIS 2-4 and MIS 6 are missing from their sites but the records plotted at the ICS web site indicate that these intervals are present. As with nearly all types of geological archives (lake sediments, ocean sediments, loess sediments), sedimentological dynamics can be highly variable site to site; missing section at one site on the Loess Plateau is not sufficient to infer

that it is missing at another 40 km distant; more direct evidence is needed prior to publication. In the absence thereof, eliminating this component of the paper might be advisable, necessitating a revision of the title and general motivation component of the paper.

II. Implications for glacial-interglacial depositional and erosional processes and mechanisms operating in this desert-marginal region

This is where the manuscript excels, strongly. The authors derive a cogent and internally consistent framework for linking changes in high-latitude ice volume to the expansion and contraction of desert margins with consequent impact on the unique sedimentation and erosional processes operating at this northern margin of the Loess Plateau and how to interpret alternating soil-sandy loess horizons. It's not easy to write text that conveys the impact of these types of dynamics on the development of a sediment section; the authors' description accomplished this with clarity, to the point that one does question the integrity of the ICS section. However, as stated above, if this aspect is the main focus of the paper – in the title and abstract, the next step needs to be taken – a reinterpretation of at least the top 300 kyrs of the type-section itself; apply the new model developed in this paper and make a direct case for inaccuracies.

III. Implications for our understanding of monsoon dynamics relative to forcing mechanisms and timing at the precession band.

This also is where the manuscript excels, strongly. This is the first concrete proof of a lag in the loess response at the precession band, an issue that has been endlessly debated and discussed because it carries with it a fundamental means of identifying the underlying physics of the monsoon response. Briefly, if northern hemisphere summer insolation were the only mechanism driving monsoon circulation, then we expect a near zero phase response. If the monsoon is equally sensitive to insolation as it is to ice-volume/carbon cycling then we expect a phase response in between the timing of max insolation and minimum ice volume. If the monsoon response significantly lags minimum ice volume then a third mechanism may be at play. The OSL dating pins the lag down, independent of previous chronologies, many of which were developed on the assumption changes in the loess grain size record were in phase with changes in the global benthic stack (grain size records across the plateau are strikingly similar in structure to the benthic stack). This work confirms the lag and establishes its magnitude, at least in this region of the plateau; it's a significant finding with significant implications, as stated in the manuscript.

In summary, and with due deference to OSL experts if required, the manuscript would be suitable for publication in Nature Communications if it included a robust assessment of inaccuracies in the ICS type section itself, based on reinterpretation using the new sedimentation/erosion model put forward in the paper.

We thank the reviewers for their extremely positive and encouraging comments on the significance and content of our manuscript and analysis, as well as their suggestions for improvements. We have followed the suggestions as closely as possible and believe these have greatly strengthened the manuscript. A point by point response to each comment is outlined below.

Reviewer #1:

(1) The highly spatially variable relationship among age, depth, and sediment type and preservation is noted. Some sections near Jingbian record deposition while others record hiatuses. No single section preserves the full sequence as shown in the composite climate record. This is a very intriguing finding. The following sentence is quoted from the manuscript: "As such, these gully sequences require consideration as dynamic landforms, where the morphological context must be taken into account, together with the stratigraphic sequence." The authors may want to consider elaborating on this statement. What kinds of landforms? Wind-parallel linear loess topography has been documented on the desert marginal Loess Plateau. Is there any correlation between the locations of the sections and the locations of modern ridges/troughs/gullies? Or perhaps the stratigraphy records paleo-topography?

> Thanks for this suggestion. To clarify this specific sentence; here we primarily mean that local gully morphology plays a key role in sediment deposition and preservation (we now clarify this by changing the text to '...where gully geomorphology and local morphological context must be taken into account,...' (lines 111-112). To address the wider question; many sections are indeed located in gully areas. The locations of these gullies seem related to wider drainage patterns as we are close to the Loess Plateau escarpment (as noted below) and are therefore at a drainage divide. There may be some relationship to ridge/trough orientation but we do not know if this is causal or rather related to the position of the escarpment, itself driven by wind erosion, as the reviewer notes. We can also confirm that in some sections stratigraphy records palaeotopography: the observation (lines 113-116) that sediments from the Holocene are preserved unconformably within the gully but not at the surface demonstrates the imprint of the gully morphology in the stratigraphic record and shows that the gully itself pre-dates these sediments. Based on the reviewer's comments we now add a sentence discussing this aspect (lines 117-118). As pedogenesis is weak at the site, the stratigraphy is relatively hard to see without independent dating, further underscoring the need for independent age assessment at these sections.

(2) Related to (1) above, it might strengthen the manuscript to briefly summarize some of the geomorphic observations which led to the hypothesis that the Loess Plateau is a dynamic landform—could even indicate some on Figure 1. e.g., Jingbian is located near the windward escarpment margin of the Loess Plateau—proposed to have retreated in the windward direction because of wind erosion during peak glacials. It is most prominent to the west, beyond the city of Dingbian, where there are also large yardangs (or yardang-like landforms) sculpted in Quaternary strata, and wind gaps along the crest of the escarpment. These observations may further strengthen/broaden the conclusions and implications of this study.

> We agree and have added a summary of some of the observations that back up the aeolian cannibalism hypothesis, and slightly extended the discussion of this aspect of the manuscript (lines 203-209, 214-216). However, we have not added the escarpment position to Figure 1 because the large scale map coverage of the figure would make this feature hard to represent in a meaningful way.

Reviewer #2:

Main text:

Line 44: strongly suggest adding the word 'terrestrial' to read "one of the world's most important TERRESTRIAL climate archives" (I'm thinking of ice-core and marine scientists reading this!)

> "terrestrial" was added.

Line 46: suggest adding a comma after the bracket to help the reader

> A comma was added.

Line 59: suggest adding a comma after "record is limited" to help the reader

> A comma was added.

Line 82-83: "This radically changes the palaeoclimatic sequence preserved at the site..." – I'm not sure that the authors really mean this as written?; it's the word 'sequence' which is potentially an issue here as the reader could confuse this with the sedimentary 'sequence', which is obviously not changed – rather, it is obviously the timescale and hence also the palaeoclimatic interpretation and the accumulation rates etc that are all changed.

> Changed to "This radically changes the palaeoclimatic interpretation of the sedimentary sequence preserved at the site" (line 82).

Line 107: suggest adding a comma after "site" to help the reader

> A comma was added.

Line 126: "The large hiatuses appear to terminate following...at the end of the MIS 2-4 and 6 glacial stages (Terminations I and II)." – is the word 'following' appropriate or potentially misleading? The end of one hiatus seems essentially coincident with termination I, whilst accumulation after another hiatus resumes before TII? This may just be an issue of phrasing, rather than an issue of interpretation of the data.

> We do not feel confident enough in the precision of our chronology to be sure that sedimentation resumes before TII rather than at or after TII. We have thus modified the text to "The large hiatuses appear to terminate close to or following the rapid shift away from peak Northern Hemisphere ice volume at the end of the MIS 2-4 and 6 glacial stages (Terminations I and II)" (lines 131-132)

Line 130/131: Difficult to comment on "full preservation of the loess record" for MIS 8 when the full stage is not captured in the record shown – however, where sediments are present (ie from ~270 ka onwards) then the records do indeed seem to be preserved. Again, probably more of a phrasing issue which can easily be resolved.

> We have modified the text to read "During MIS 7 and second half of MIS 8 there is..."

Line 152/153: "Sand was deposited at the end of both hiatuses, indicating both an expansion of the Mu Us desert and some dune stability." Could enhanced/sustained wind competency also potentially

be a factor here? Please comment on whether/how one might distinguish between these various factors (sediment availability, sediment trapping, and wind strength)?

> We agree that factors such as wind strength, trapping (and stabilization), availability etc. could all impact the expansion of sand activity and deposition at the site, and indeed note this further on in the manuscript in discussion of these results (lines 181-185). However, beyond the overall causal mechanism relating to ice volume driven expansion of the Siberian High, we are reluctant to specify the local process cause of this enhanced/expanded sand activity as this is not something that can be easily derived from our sediment record. A number of papers deal extensively with this complex topic using modeling, theory and proxy data (e.g., Tsoar, 2005) and a proper discussion of how to distinguish between such factors would be beyond the scope and focus of this submission. In any case, the preserved sand units denote expanded Mu Us sand dunes over the site, and all of the factors above may be driven by an enhanced Siberian High, the main message of our manuscript.

Sentence from line 163-166: *"In contrast to traditional...we propose a different model for Jingbian."* This phrasing is very awkward for the reader; it is difficult to read due to the many clauses that appear before the main points which start-and-finish the sentence (and which are highlighted in the quoted text pulled out at the start of this comment) i.e. the main point is that the authors are proposing a different model to the traditional view, but this gets lost due to the current sentence structure. Suggest rephrasing this important sentence, for clarity.

> Text has been modified to read "In traditional Loess Plateau chronostratigraphic models, loess / sand units and palaeosols are considered of glacial and interglacial age, respectively. Here, we propose a different model for Jingbian. In our view, palaeosol units are indeed indicative of interglacial phases.... However, ... stadials within interglacials..." (lines 168-171)

Sentence on line 175-180: *this is a long yet densely packed sentence; suggest that the authors revisit this sentence and check that it conveys their ideas as clearly as possible.*

> Text modified to: "Although the proxy records show general antiphase behaviour of the EASM with the EAWM (Fig. 3), sand accumulation can also occur even during enhanced summer monsoon conditions (e.g., MIS 5e). This suggests that (i) sediment availability, (ii) EAWM/Siberian High driven winter aridity, and (iii) cold air outbreaks drive dune mobility, desert expansion and sand deposition at desert marginal sites¹⁵. This is in contrast to the idea that expansion and deposition is controlled by summer monsoon-driven moisture availability¹⁰." (lines 180-186)

Line 178: *"suggesting that sand sediment availability..." – the word 'sand' or 'sediment' is perhaps intended here, but presumably not both?*

> see above

Sentence on line 202-205: *"That this reworking at Jingbian only occurs during those glacial phases (the most recent) with greatest ice volume is consistent with a long-term increase in aeolian accumulation rates over the Quaternary" – I agree with the sentiment of this sentence and suspect that the authors are correct in what they say, but it is difficult to draw this conclusion solely from the data in this paper (which is what the sentence implies as it is current written), because the 2 reworking/hiatus events take place in the only 2 full glacial times identified and dated in the sequence (MIS 2-4 and MIS 6); for MIS 8 and beyond no record is shown beyond ~270 ka. Hence, it is not possible to comment on the presence or absence of materials from the sedimentary record (i.e. on hiatuses) for stage 8 and earlier because no records are shown for these glacial periods; additionally, there is no independent*

numerical chronology for such earlier glacial stages (prior to ~270 ka) hence it is difficult to envisage how any hiatus would be identified in such a case.

> We have removed “only” in the first part of this sentence (line 216)

Line 220: add a hyphen between “EASM” and “driven” to read “EASM-driven peaks preserved in...”

> A hyphen was added between “EASM” and “driven”.

Line 223: “with a consistent lag...” – it’s not a ‘consistent lag’ (which could be taken to imply a constant temporal offset), but it does ‘consistently lag behind July insolation’ (in the sense that it is always behind rather than in front of or consistent with the timing of July insolation)

> Text changed to “but systematically lags behind July insolation...” (line 236)

Line 228: suggest adding a comma after ‘precession’ to help the reader

> A comma was added.

Line 229: suggesting adding a comma after ‘response’ to help the reader

> A comma was added.

Line 238: add units (ka) to the figure of “4.9”

> “ka” was added after “4.9”.

Text within and around lines 249-254: could it also be the case that significantly enhanced sand content (whatever the cause of this may be) simply precludes formation of a palaeosol/development of enhanced magnetic susceptibility ?

> No, we don’t think so because we have an increased sand content in the well-preserved MIS5e palaeosol, in which magnetic susceptibility is also enhanced.

Line 253: either ‘these data’ or ‘this dataset’ (but not ‘this data’)

>Corrected to “this dataset”

Sentence around line 238 and line256 etc regarding changes in the East Asian Summer Monsoon (EASM) record preserved at Jingbian being reported to lag insolation changes by up to ~4.5ka: I suspect that the authors are correct, but it should also be noted that this lag typically falls within the 1-2 sigma uncertainties on the individual ages and/or the age model, typically used to assess whether or not events are truly separated in time. Having said this, on the positive side, there is a consistency in the data in that a lag is always detected in each case, which suggests that (as long as there is no systematic issue in the OSL dating affecting all ages, such as assessment of the water content) then there may well be a genuine lag in the MS record that is being preserved at Jingbian. The authors may wish to address this issue of uncertainties and the ability to discern these apparent lags in the record.

> Interesting point raised by the reviewer. Indeed, the assessment of the appropriate life-time average water content and its associated uncertainty is important in this study.

To address this point we here first discuss our choice of water content and its uncertainty with respect to literature values and its relevance to individual samples using the section containing the

Holocene soil (section D). We then investigate the dependence on different water content assumptions of the apparent lag between our luminescence dated MS record and the insolation record.

Firstly, although there is some variability in the published water content values for Chinese loess (see discussion in Stevens et al., 2013), the values used in this study, of $15\pm 5\%$ w.c. for soil and $10\pm 5\%$ w.c. for loess layers, are in line with previous water content assessments for loess/palaeosols from sites in the N and NW of the Chinese Loess Plateau (e.g. Buylaert et al., 2008; Sun et al., 2010). In addition, Chen et al. (2015) used a value of $10\pm 5\%$ for a single sample collected in the S8 palaeosol at Jingbian.

We next consider the water content required to reduce the EASM lag to 0 ka for section D. For the two Holocene samples (D38132, w.c. 15% and D38136, w.c. 10%) this would require increasing the water content to $\sim 30\%$ and $\sim 25\%$, respectively. These water contents are 3 standard deviations from the values used and are close to saturation for sandy loess deposits. However, it is likely that the upper loess-palaeosol units at Jingbian have been well-drained since deposition: (i) the gully is at least 10 ka old since the Holocene soil is inset into the gully system. (ii) The current water table is now around 300 m below the sampling level in a >280 m deep gully system (Ding et al., 2005). The river into which the gully flows has incised into Pliocene red clay below the Quaternary loess. The age of this feature is unknown but is likely to be at least multiple glacial-interglacial cycles. It is thus expected that the upper loess-palaeosol units at the site have remained at least several tens of m above the water table for the majority and probably all of their burial life-time. Thus, we consider it unlikely that the life-time average water content of this site can approach the levels that would be required to reduce the lag to 0 ka. It is also worth noting that the water content values required for a zero lag would exceed almost all published values for even southern Chinese Loess Plateau sites, where precipitation levels are double those at Jingbian. If, on the other hand, our water content estimates are too high, the dose rates would be higher, the luminescence ages lower and the lag with the insolation larger. Thus, in all likely water content scenarios there is a consistent lag between the EASM recorded in loess and insolation.

If we now make the additional assumption that the underlying mechanisms causing the insolation lag have not varied systematically with time (which ought to be safe given the lack of an obvious systematic change in ice volume and CO_2 back in time at insolation inflection points), we would in turn expect the insolation lag to have remained constant within some bounds over the past ~ 250 ka. This is precisely what is observed in our data (Supplementary Table 3 and black symbols in new Supplementary Fig. 7). However, increasing the water content by one standard deviation (i.e. from 15% to 20% for soil and from 10% to 15% for loess) causes an increase of 4.7% and 4.1% in the quartz and feldspar ages, respectively. Recalculation of the insolation lag using ages based on these higher water contents introduces a negative trend in the insolation lag versus insolation inflection time point graph (red symbols in new Supplementary Fig. 7). Indeed, using these water contents suggests the physically unrealistic scenario that prior to 130 ka the loess record of monsoon variability formed *before* the change occurred in the driving force (change in insolation). A similar but positive trend in the size of the lag is observed when the water content is decreased by 5% (green symbols in new Supplementary Fig. 7). We conclude that our current water content assumption remains the most likely. It does not produce any systematic trend in the insolation lag with time and, if anything, the uncertainty on the water content has been overestimated.

Because this is an important point raised by the reviewer we expand on this in Supplementary Note 3 "Dependence of luminescence ages on water content". In this section the three main arguments are:

- (i) chosen values are in line with those used in the literature
- (ii) the site is well-drained and likely to have been so throughout the period of interest
- (iii) the insolation lag sensitivity analysis (Suppl. Fig. 7) outlined above further supports our chosen values

In the main text of the manuscript we now also specifically refer to the water content assumption on lines 95-96

Line 268: add an Oxford comma after 'simulations' to help the reader
> A comma was added.

Line 296: consider adding a comma after 'CLP' to help the reader
> A comma was added.

Line 312: separating "K-feldspar rich extracts" isn't necessarily guaranteed from using a single density separation at 2.58 g/cm³ – please state the K content (as %K or %K₂O) here to qualify/demonstrate that the fractions really are "K-feldspar rich" or moderate the text.
> We have added "(K content = 12.70±0.10%, n = 5)" to the text on line 326

Line 316: please define, within the text, what "absence of a significant IRSL signal" is eg was this assessed using an IR-OSL depletion ratio, or on the basis of raw IRSL counts? If the latter, then how does the raw IRSL count translate into an impact (or not) on a blue OSL signal used for dating?
> The OSL-IR depletion ratio (Duller, 2003) was used to assess the purity of the quartz extracts. This has been added to the text (lines 329-331).

We have also investigated the effect of the OSL IR depletion ratio and the standard blue OSL recycling ratio on the quartz D_e value. No relationship can be found between the D_e and the OSL IR depletion ratio (or blue recycling ratio). These data are now presented as a new Supplementary Figure 3 and are also discussed in the Supplementary Note 2.

Line 325: state filter thicknesses (mm)
> Filter thicknesses was added to the text.

Line 325-326: amend text to reflect the fact that 2 protocols were used (one for quartz, and one for feldspars) – at present it sounds like everything was measured using one single sequence e.g. amend to say "Single aliquot regenerative dose protocols (M&W, 2000) were used..."
> Complied. Text has been changed according to referee's suggestion (lines 342-343).

Line 337: "325°C" for how long (s)? Please add to the text as all other stimulation times are quoted here.
> Duration of clean-out has been added to the text (line 355).

Line 559-564: caption could be improved here to help the reader e.g. adding line colours to the caption text where records are mentioned in the text, and also adding individual graph names (a-f) to

the caption where relevant (eg to denote which groups of figs are related to which specific MIS stages named in the caption). Please also state in the caption whether the red data points are the calculated luminescence ages, or whether they are the ages after a Bayesian fit to the whole luminescence dataset at any given section

> Caption corrected accordingly. The red data points represent values of measured magnetic susceptibility plotted against the age, so these are not luminescence ages. This is now clarified in the revised caption.

Fig 3, within the main manuscript: it could potentially be helpful (and very impressive!) to consider adding an indication of the number and location of independently dated luminescence samples to Fig 3 e.g. by showing a dot for each luminescence age generated in a single line just offset above the age-axis of Fig 3, to show the location (in time, rather than in depth) of each luminescence age determination.

> Luminescence ages were added to Fig 3. (see revised figure, panel f).

It would also be helpful to add some stratigraphic details to Fig 3, or at least to indicate the location of the major loess units and palaeosols as defined (presumably) on the basis of magnetic susceptibility, if not logged/visible in the field (e.g. to support arguments/observations made in the text, such as those ~line 170).

> We did not comply with this suggestion from the referee because the focus of Fig. 3 is not the stratigraphy and we risk overloading an already busy figure (now that also the individual ages have been added at the bottom). The stratigraphic information (magnetic susceptibility and loess (L) and soil (S) nomenclature) is available on Fig. 2. In any case, the main loess, palaeosol layers can be distinguished in Fig. 3 as well, based on the magnetic susceptibility curve in panel d.

Supplementary Information:

Line 84: please comment on why small test doses should be avoided (e.g. carry-over of charge?) and give refs where appropriate (e.g. Yi et al 2016; Colarossi et al, 2017).

> Indeed, this may be due to carry-over of charge although we have not explicitly tested this. We have added the following sentence to Supplementary Note 2, section "Feldspar post-IR IRSL signal": "Colarossi et al. (2017) have shown that in their sample at least part of this effect could be attributed to charge carry-over from L_x to T_x ."

Lines 94-95: please just clarify by stating in the text the bleaching conditions where >80 days and ~300 h are mentioned (eg daylight bleaching? Sol2? etc).

> Clarified in the text. We added "in a Hönle SOL2 solar simulator with a lamp-sample distance of 80 cm" at the end of the sentence.

Add a comma to line 97 to make it easier for the reader (eg after the sample name?).

>Complied, a comma was added.

Line ~122 and Fig S4: please state how the K contents are obtained as Kook et al. 2012 is just a conference abstract and doesn't contain these details. The XRF attachment gives a relative assessment of the K, Na, Ca composition (as subsequently expressed in Fig S4); please explain in the SI text how concentrations (as % K or %K₂O) are then derived. Was any other assessment of the K-

content of the separates undertaken (eg beta counting of the material used for dating), and if so what values were obtained?

> No other analyses were undertaken. The following text has been added to Supplementary Note 2, section "Dose rate and luminescence ages":

"After chemical separation we are confident that our samples are almost entirely made up of quartz and feldspar. Thus, the XRF instrument is calibrated using a set of standards which are notionally identical, in terms of composition, to end members of the alkali- and plagioclase feldspar series and to quartz; these standards are arranged to fully cover the sample area. This allows us to convert our count rates under the Na, K and Ca X-ray peaks into relative feldspar contributions (i.e. % of total made up of K-feldspar, etc.). The calibration further allows us to attribute a proportion of Si counts to the 3 feldspar contributions, and any remaining Si counts are attributed to quartz. In general, the sum of the 4 components will be less than unity because the sample area may not be fully covered, and so all contributions are normalized to 100%. Once the feldspar analyses have been located on the ternary diagram the results can be converted to absolute concentrations of K (and Na and Ca if desired) using stoichiometry."

Fig S2: please state in the caption the number of datapoints that underlie each datapoint plotted in Figs a, b and d (eg the mean and standard deviation of 3 datapoints? etc). I think that the data shown in Figs c and d are derived from the same aliquots, in which case it would be good to say this in the figure caption.

> This information has been added to the caption.

Fig S3 caption: the D_e value quoted in the caption for sample D38135 (47.8 ± 0.6 Gy) is different to the value in the SI text (43 ± 2 Gy) – please amend/explain. A comment is also made in the caption that the test dose varies by 40-80 % - please state whether this variation in test dose is just relatively random, or whether it is due to using one (or a limited number) of fixed test dose values, or whether it was varied systematically with increased expected D_e , etc.

> The referee is correct that the D_e value quoted in the Supplementary Table 2 is different from the one in the original Fig. S3. This is because in the Supplementary Table 2, the quoted D_e values have been corrected for a residual dose of 5 ± 2 Gy ($47.8 \pm 0.6 - 5 \pm 2$ Gy gives 43 ± 2 Gy). In the caption of Supplementary Figure 4 we have added that the D_e value is not corrected for residual dose (because this is a natural + beta dose recovery test).

Information about the test dose variation has been added to the caption.

Note also that the last line of the caption should read "the data ARE consistent" (or "the dataset is consistent"), not "the data IS consistent"

> Corrected

References in reply to reviewer 2:

Buylaert, J.P., Murray, A.S., Vandenberghe, D., Vriend, M., De Corte, F., Van den haute, P., 2008
Optical dating of Chinese loess using sand-sized quartz: Establishing a time frame for Late Pleistocene climate changes in the western part of the Chinese Loess Plateau. *Quaternary Geochronology*, 3, 99-113.

Chen, Y., Li, S.-H., Li, B., Hao, Q., Sun, J., 2015. Maximum age limitation in luminescence dating of Chinese loess using the multiple-aliquot MET-pIRIR signals from K-feldspar. *Quaternary Geochronology* 30, 207-212.

Colarossi, D., Duller, G.A.T., Roberts, H.M., 2017. Exploring the behaviour of luminescence signals from feldspars: Implications for the single aliquot regenerative dose protocol. *Radiation Measurements*, in press <https://doi.org/10.1016/j.radmeas.2017.07.005>

Duller, G.A.T., 2003. Distinguishing quartz and feldspar in single grain luminescence measurements. *Radiation Measurements* 37, 161–165.

Stevens, T., Adamiec, G., Bird, A.F., Lu, H., 2013. An abrupt shift in dust source on the Chinese Loess Plateau revealed through high sampling resolution OSL dating. *Quaternary Science Reviews* 82, 121-132.

Tsoar, H., 2005. Sand dunes mobility and stability in relation to climate. *Physica A: Statistical Mechanics and its Applications* 357, 50-56.

Sun, Y., Wang, X., Liu, Q., Clemens, S.C., 2010. Impacts of post-depositional processes on rapid monsoon signals recorded by the last glacial loess deposits of northern China. *Earth and Planetary Science Letters*, 289, 171-179.

Reviewer #3:

The authors make a strong case for a significant amount of disturbance and reworking, resulting in well-documented hiatuses labeled in figure 2. Beyond these labeled features, the authors interpret the sections as largely intact and continuous. In the case of Section B (Figure S5_B), however, the OSL dates appear to indicate, if taken at face value, some larger-scale age reversals or disturbed section. See, for example the intervals ~500 to 600 cm, 600 to 800 cm, and 800 to 900 cm where age decreases with depth in the section. Might these indicate slumping with possibly repeated section or simply disturbed section? If so, such disturbance may be why the structure in figure 3d does not match that in the Baikal record within MIS 7, unlike the reasonably good match in MIS 5. In any case, the authors point is valid – that this desert-margin location appears prone to strong depositional/erosional dynamics and may not be the strongest choice for locating an ICS benchmark stratigraphic section.

> We agree with the reviewer that in the quoted depth intervals of section B the ages are quite scattered and this was also acknowledged in the main text (end of second paragraph under section “EASM, ice volume and lagged response to insolation forcing”; lines 265-268). As the reviewer suggests, there may be undetected slumping or other disturbance in this part of the section although we are more inclined to attribute the apparent inversions to undetected systematic dating errors that occur close to the limit of the range of the feldspar dose response curve. We also hypothesized in the original text that there may be an undetected erosional event that has removed a magnetic susceptibility peak in MIS7. However, any discussion of disturbance and/or dating error would be speculative and we have chosen not to make any corrections to the text.

We are pleased to read that the reviewer agrees with our statement in the main text that the Jingbian site may not be a suitable ICS benchmark stratigraphic section.

Such a general statement, in and of itself, is valid but the finding is then extrapolated to the ICS type section 40 km to the NW with the statement (abstract lines 21-24, title, text lines 194-194) that the chronology at the type section is inaccurate. A statement of this order should be backed up with an in-depth comparison documenting where, in the last 300 kyrs, the chronology of the type section is compromised. For example, the authors document clearly that glacial intervals MIS 2-4 and MIS 6 are missing from their sites but the records plotted at the ICS web site indicate that these intervals are present. As with nearly all types of geological archives (lake sediments, ocean sediments, loess

sediments), sedimentological dynamics can be highly variable site to site; missing section at one site on the Loess Plateau is not sufficient to infer that it is missing at another 40 km distant; more direct evidence is needed prior to publication. In the absence thereof, eliminating this component of the paper might be advisable, necessitating a revision of the title and general motivation component of the paper.

> We wish to emphasize here that we have not extrapolated our results to the ICS type section 40 km to the NW. Our work was carried out at precisely the ICS type section. Unfortunately, some confusion has arisen because the coordinates published in Ding et al. (2005) and hence on the ICS chart are incorrect. It appears that we have not demonstrated clearly enough in our submitted manuscript that we are in fact working at the ICS site, given this previous error in past publication. We take the opportunity here to remove any doubt about this and are grateful to the reviewer for raising this so we can make this as clear as possible in the manuscript. There are several lines of evidence that our Jingbian site is the ICS type section.

- 1) We have received the GPS coordinates of the Jingbian section from one of the co-authors (E. Derbyshire) on the Ding et al. (2005) paper. (Derbyshire E., personal communication 2015). These coordinates (37°29'58.74" N, 108°54'2.72" E) refer to a pylon/mast about 100 m from the gully containing our sections. Our coordinates for section A (37°29'52.8"N, 108°54'14.4"E) are ~330m from the location given by Derbyshire (see also Supplementary Figure 1).
- 2) In Ding et al. (2005) the site is described as *"The Jingbian section (37°40'54" N, 108°31'15" E), at 1370 m above sea level, lies on the summit of the Baiyu Mountains. It is located only ~12 km south of the present margin of the Mu Us Desert. No local sources of sand, such as river channels, are present in this area"*. From Google Earth we can see that this description fits very well with our site location (and the coordinates above) but does not fit with the coordinates given in Ding et al. (2005). Their coordinates point to a location well within the Mu Us desert, surrounded by sand dunes and only 2.7 km from the Hongliu River.
- 3) In Ding et al. (1999) it is stated that *"The Jingbian section is located at Guojialiang"*. The coordinates given by Derbyshire (pers. comm., 2015) are ~100 m from Guojialiang and our coordinates for section A ~350 m from Guojialiang. In contrast, the coordinates given by Ding et al. (2005) are 39.1 km from Guojialiang, despite the fact that this publication also refers to the Ding et al. (1999) article. Thus this must be an error in the coordinates cited in Ding et al. (2005).

In summary, we are certain that we were working at the same site as Ding et al. (1999, 2005) and hence that we were working at the ICS type section. However, we accept that this was not clear from the previous version, given the error in the coordinates cited in Ding et al. (2005). As such, we have modified the following sentence in the main text to make this clear to the reader *"Our new luminescence age model is based on 220 ages on samples taken with a vertical spacing of between 5 and 40 cm at 5 Jingbian sections dug at the ICS stratotype location"* (lines 91-92). We have also updated the Supplementary Note 1 - Study site and this now contains a full description of the evidence for our working at the ICS section noted above.

Thus, our description of the general motivation for our paper remains unchanged and as a result we have not modified the title.

It's not easy to write text that conveys the impact of these types of dynamics on the development of a sediment section; the authors' description accomplished this with clarity, to the point that one does question the integrity of the ICS section. However, as stated above, if this aspect is the main focus of the paper – in the title and abstract, the next step needs to be taken – a reinterpretation of at least

the top 300 kyrs of the type-section itself; apply the new model developed in this paper and make a direct case for inaccuracies.

> We are pleased to read that the reviewer finds our arguments convincing and clear. Given that we are confident that we were working at the ICS section (see reply to previous comment) this aspect of the main text has not been changed.

References in reply to reviewer 3:

Derbyshire, E., 2015. Personal communication (available upon request from the corresponding author).

Ding, Z., Sun, J., Liu, T., 1999. Stepwise advance of the Mu Us desert since late Pliocene: evidence from a red clay-loess record. Chinese Science Bulletin 44, 1211–1214.

Ding, Z.L., Derbyshire, E., Yang, S.L., Sun J.M., Liu T.S., 2005. Stepwise expansion of desert environment across northern China in the past 3.5 Ma and implications for monsoon evolution. Earth and Planetary Science Letters 237, 45-55.

REVIEWERS' COMMENTS:

Reviewer #1 (Remarks to the Author):

I have read the authors' response to reviewers and the significantly improved / more clear revised manuscript. My comments were appropriately addressed by the authors and my assessment is that the manuscript is suitable for publication in its present form. It will be an excellent contribution in Nature Communications.

Sincerely,

Paul Kapp
30 November 2017

Reviewer #2 (Remarks to the Author):

Reviewer #2:

I am content with the responses given and actions taken by the authors in light of reviewers' comments. Speaking to the specific comments I have raised, I believe that the additional information given in the paper in response to these comments has enhanced the clarity and transparency of the work, and illustrates to the reader the careful nature of the data acquisition and the thoughtful subsequent interpretation of the data.

I find this to be an important, internationally-significant paper, based on high-quality numerical age data in unprecedented abundance, and as such believe it will be highly appropriate for publication within the proposed journal.

Reviewer #3 (Remarks to the Author):

Author revisions to the main text and Supplementary Note 1 - Study Site, make it clear that they are indeed working at the same location as the ICS stratotype site, clearly resolving the only concern I had with this manuscript being suitable for publication in Nature Communications. The manuscript makes a clear and concise case that the ICS stratotype site is incomplete, provides mechanistic interpretation of sedimentation processes at this desert-margin location, and addresses the precession-band phasing of the monsoon response to insolation forcing. Thus, the work is of interest to a broad array of readers and meets the threshold for publication in Nature Communications.

We thank the reviewers for their extremely positive and encouraging comments on the significance and content of our improved manuscript and analysis.

Reviewer #1:

I have read the authors' response to reviewers and the significantly improved / more clear revised manuscript. My comments were appropriately addressed by the authors and my assessment is that the manuscript is suitable for publication in its present form. It will be an excellent contribution in Nature Communications.

Sincerely,

Paul Kapp

30 November 2017

Reviewer #2:

I am content with the responses given and actions taken by the authors in light of reviewers' comments. Speaking to the specific comments I have raised, I believe that the additional information given in the paper in response to these comments has enhanced the clarity and transparency of the work, and illustrates to the reader the careful nature of the data acquisition and the thoughtful subsequent interpretation of the data.

I find this to be an important, internationally-significant paper, based on high-quality numerical age data in unprecedented abundance, and as such believe it will be highly appropriate for publication within the proposed journal.

Reviewer #3:

Author revisions to the main text and Supplementary Note 1 - Study Site, make it clear that they are indeed working at the same location as the ICS stratotype site, clearly resolving the only concern I had with this manuscript being suitable for publication in Nature Communications. The manuscript makes a clear and concise case that the ICS stratotype site is incomplete, provides mechanistic interpretation of sedimentation processes at this desert-margin location, and addresses the precession-band phasing of the monsoon response to insolation forcing. Thus, the work is of interest to a broad array of readers and meets the threshold for publication in Nature Communications.